# Formal Methods in Robot Policy Learning and Verification: A Survey on Current Techniques and Future Directions

**Anastasios Manganaris**                    *amangana@purdue.edu*
*Department of Computer Science*
*Purdue University*

**Vittorio Giammarino**                    *vgiammar@purdue.edu*
*Department of Computer Science*
*Purdue University*

**Ahmed H. Qureshi**                    *ahqureshi@purdue.edu*
*Department of Computer Science*
*Purdue University*

**Suresh Jagannathan**                    *suresh@cs.purdue.edu*
*Department of Computer Science*
*Purdue University*

**Reviewed on OpenReview:** *https://openreview.net/forum?id=DZkikdg5sl*

## Abstract

As hardware and software systems have grown in complexity, Formal Methods (FMs) have been indispensable tools for rigorously specifying acceptable behaviors, synthesizing programs to meet these specifications, and validating the correctness of existing programs. In the field of robotics, a similar trend of rising complexity has emerged, driven in large part by the adoption of Deep Learning (DL). While this shift has enabled the development of highly performant robot policies, their implementation as deep Neural Networks (NNs) has posed challenges to traditional formal analysis, leading to models that are inflexible, fragile, and difficult to interpret. In response, the robotics community has introduced new formal and semi-formal methods to support the precise specification of complex objectives, guide the learning process to achieve them, and enable the verification of learned policies against them. In this survey, we provide a comprehensive overview of how FMs have been used in recent robot learning research. We organize our discussion around two pillars: policy learning and policy verification. For both, we highlight representative techniques, compare their scalability and expressiveness, and summarize how they contribute to meaningfully improving realistic robot safety and correctness. We conclude with a discussion of remaining obstacles for achieving that goal and promising directions for advancing FMs in robot learning.

## 1 Introduction

Both in natural and artificial settings, it has been repeatedly observed that surprising levels of complexity and capability can *emerge* from systems built by massively scaling and composing simple components (Anderson, 1972). The "bitter lesson" learned by artificial intelligence researchers is one instance of this phenomenon: methods that scale effectively with data and compute tend to outperform those that rely on explicitly incorporating human knowledge (Sutton, 2019). The emergent complexity of such scalable systems is one of the most significant factors behind the success of DL (Kaplan et al., 2020). When combined with well-developed theoretical frameworks for general decision-making and control, DL techniques have scaled

from achieving superhuman performance in game playing (Mnih et al., 2015), to enabling effective robotic manipulation (Levine et al., 2016; 2018), and now to powering increasingly general embodied agents and foundational models for robotics (Reed et al., 2022; Hu et al., 2024; Firoozi et al., 2024). With this flexibility, it is becoming viable to deploy robots alongside humans in unstructured environments and with open-ended goals—fulfilling roles such as household assistants (1X Technologies, 2025), autonomous vehicle controllers (Favaro et al., 2023), and medical assistants (Empleo et al., 2025).

However, these new use cases demand higher levels of confidence in deployed robot policies and, consequently, greater levels of policy interpretability, robustness to uncertainty, and compliance with behavioral and safety specifications. Alarmingly, achieving such confidence appears to be fundamentally constrained by the very increase in policy complexity that has enabled these systems' viability. Learned robot policies, typically parameterized by deep NNs, are well known for their lack of interpretability (Zhang et al., 2021), poor generalization to out-of-distribution settings (Ross & Bagnell, 2010; Park et al., 2024), and vulnerability to adversarial inputs (Szegedy et al., 2014; Xiong & Jagannathan, 2024). These limitations are particularly concerning in safety-critical or high-stakes environments, where failures can lead to catastrophic outcomes. Conventional approaches to specifying robot behavior, such as reward functions or behavioral demonstrations, have so far failed to address these issues of policy trustworthiness. Without any guarantees, they can lead to unsafe, under-performing, or misaligned behaviors due to phenomena such as reward hacking in Reinforcement Learning (RL) (Skalse et al., 2022) or overfitting in imitation learning (Goodfellow et al., 2016). Even when safety is not the primary concern, these approaches also lack the expressive power to formally and concisely define complex, compositional, or temporally extended tasks.

In the world of software, similar safety concerns emerged as increasingly complex programs were deployed in safety-critical settings (Baase, 2008), and these concerns have motivated the development of Formal Methods (FMs) to mitigate or eliminate these risks. Central to these and subsequent FM frameworks is the capability to rigorously specify system behavior, for which modal logics like Linear-time Temporal Logic (LTL) (Pnueli, 1977) have been used with great success. After formally specifying some behavior, FMs can allow a user to directly **synthesize** high-level, correct-by-construction programs from a Formal Specification (FS) (Church, 1963; Solar-Lezama, 2009; Srivastava et al., 2010). Closely related FMs, such as Model-Checking, provide ways to formally **verify** that complex programs satisfy their specifications (Baier et al., 2008; Kästner et al., 2018; Murray et al., 2013). Formal specification, synthesis, and verification now constitute a complete toolkit for developing highly dependable software systems.

In robotics, these FMs have also been applied and extended to ensure the correctness and safety of controllers *without* any learning components. For example, specification languages such as Signal Temporal Logic (STL) have been developed to express requirements for continuously-evolving, real-valued properties of dynamical systems (Maler & Nickovic, 2004), and probabilistic and hybrid-system model-checking techniques have been introduced as well (Alur et al., 1991). These advancements have enabled the synthesis and verification of robotic policies based on traditional planning algorithms (Kress-Gazit et al., 2009), as well as controllers derived from optimization (Sun et al., 2022b) and control theory (Abate et al., 2019). Such formal approaches offer guarantees not only for immediate control actions but also for long-term behavioral trajectories, directly addressing the predictability and robustness of robotic systems.

With the rise of *learning*-based methods in robotics, more recent research efforts have sought to use FMs to address the critical weaknesses of purely learning-based approaches mentioned above. These new FMs have the potential to systematically provide interpretability, behavioral guarantees, and safety assurances for NN-based robot policies. Moreover, specifications used in FMs allow for more expressive and more concise behavior definitions than reward functions or demonstrations, easily capturing complex, compositional, and temporally extended requirements. This combination promises not only robust performance but also strict conformance to intended behaviors and reduced susceptibility to adversarial exploitation. Consequently, FMs represent a promising approach for transforming learned robot policies from impressive but opaque models into transparent, reliable, and trustworthy solutions ready for real-world deployment. In this survey, we contribute a thorough overview of these recently developed methods.

**Scope and Contribution**   This survey aims to provide for robotics researchers an accessible introduction and comprehensive overview for recent work at the intersection of FMs and DL for robot control. Unlike

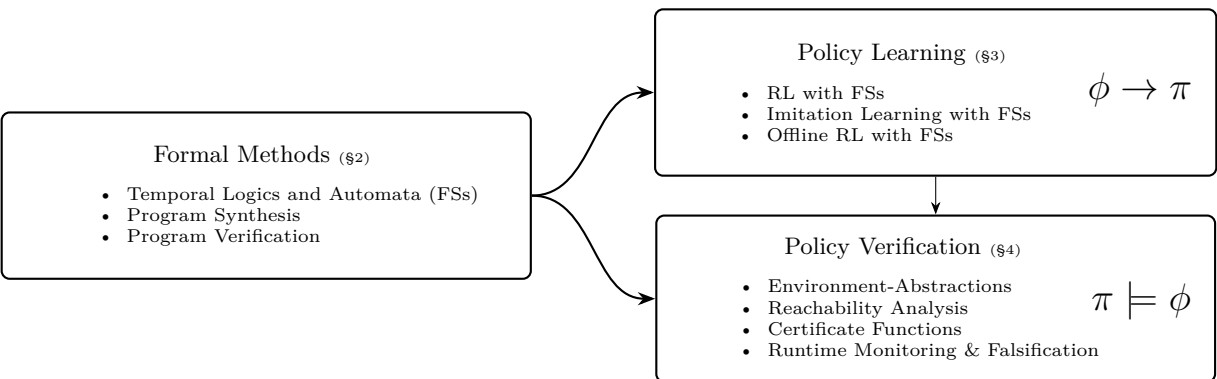

Figure 1: Our survey first provides background information on FMs and their requisite specifications, denoted by $\phi$. We then survey recent work on FMs in robotics in two categories: work on FM-informed **learning** of robot policies ($\pi$), and work on **verifying** that learned robot policies satisfy specifications ($\pi \models \phi$).

prior surveys that focus on applying learning methods to FMs (e.g., Wang et al. 2020), we instead focus on the application of FMs to learning-based methods. Specifically, this is a survey of FMs as applied to learning methods for robot policies, rather than to general Machine Learning (ML) systems (Larsen et al., 2022; Urban & Miné, 2021).

Several surveys have reviewed the use of FMs in robotics. For instance, Luckcuck et al. (2019) examine a range of FMs applications in autonomous robotic systems, with a focus on model checking for robotics software and specification languages. However, since that survey, DL-based systems have gained greater prominence in robotics and remain largely unexamined in that context. Similarly, Belta & Sadraddini (2019) survey formal synthesis methods for optimization-based robot controllers but does not address learned controllers. Some existing surveys have examined subfields of FMs relevant to ensuring the safety of learned controllers—such as learning certificate functions (e.g., Lyapunov functions, barrier functions, and contraction metrics) (Dawson et al., 2023), set-propagation-based reachability analysis (Althoff et al., 2021), Hamilton-Jacobi (HJ) reachability analysis (Bansal et al., 2017), cyber-physical system verification (Tran et al., 2022), and adjacent safe learning methods (Brunke et al., 2022). We also include the most relevant above subfields in this survey, provide an up-to-date overview of each, and refer readers to existing works for more in-depth details where appropriate. To our knowledge, no prior survey has provided such an integrated perspective on how FMs have been developed and adapted specifically for DL-based robot policy learning, and we expect this to be a valuable resource for guiding future research in this area.

**Outline** We consider the use of FMs in robot learning to be naturally summarized with the three stage flowchart shown in Figure 1. The root of the flowchart represents the existing work on FMs, from which existing techniques for obtaining and representing FSs are used as a starting point for every other FM. The next nodes representing learning a policy informed by specifications and the verification of learned policy against specifications. We therefore have adopted the same structure to organize our survey. Section 2 covers the preliminaries for understanding FMs and the existing methods of constructing Formal Specifications (FSs) which are used in the more recent work applied to robotics. Section 3 surveys methods for informing policy learning techniques with behavioral specifications to enable more complex tasks, accelerate learning, or improve safety. Section 4 surveys methods focused on verifying that learned policies conform to specifications. We clarify for each method the underlying verification approach, the specifications they support, and the assumptions they make on the environment and policy. Finally, Section 5 discusses notable gaps in the existing work to clarify directions for future work, and Section 6 concludes the survey with a summary of the above content.

## 2 Preliminaries for Formal Methods and Specifications

In this section, we introduce the fundamentals of FMs to provide background for the methods discussed in our survey. We first describe the standard model used in FMs for describing the potential behavior of discrete systems. We then describe the most widely used representations of FSs that describe behaviors in that model. Methods for formal policy learning and verification should operate with a precise description of the behavioral requirements, so these specifications serve as the starting point for every roboticist seeking to apply FMs. We conclude with a brief summary of formal synthesis and verification, which have also influenced the development of FMs in robotics.

FMs aim to ensure that systems behave as desired (Baier et al., 2008), but behaving "as desired" is fraught with ambiguity. The first step in eliminating that ambiguity is considering the system under an appropriate formal model, and the classical choice is a discrete transition system. We provide an illustrative example of such a system and an FS of complex and temporally extended behavior for the system in Figure 2. In general, a discrete transition system is a tuple $(S, A, \longrightarrow, I, AP, L)$, where $S$ is a set of states, $A$ is a set of actions, $\longrightarrow \subseteq S \times A \times S$ is a transition relation (i.e., a set of possible transitions $(s, a, s')$ between pairs of states $s, s' \in S$ given actions $a \in A$), $I \subseteq S$ is a set of initial states, $AP$ is a set of atomic propositions, and $L : S \to 2^{AP}$ is a labeling function that maps each state to the set of true propositions for that state (Baier et al., 2008). Under this model, as well as any other model that supports a similar notion of states and propositional labeling, the behavior of a system can be formally defined as a (possibly infinite) sequence of states $(s_0, s_1, \dots) \in S^\infty$, where $S^\infty$ denotes the set of all possible finite and infinite sequences over $S$. The possible sequences are further restricted to those starting with states $s_0 \in I$, and those that can be induced by another sequence of actions $(a_0, a_1, \dots) \in A^\infty$ under the system's transition relation $\longrightarrow$. As behaviors under this model are formally sequences of states, it is natural to use tools from formal languages to express specifications of desirable behavior. While there are many such tools with the expressiveness necessary to describe most behaviors relevant to robotics applications, we introduce here two categories of particularly useful and relevant formalisms: Temporal Logics and Finite State Automata (FSAs).

**Temporal Logic** LTL (Pnueli, 1977) is an extension of propositional logic. Logical expressions for some system, typically denoted by $\phi$ and belonging to a set of expressions $\Phi$, are defined over the atomic propositions ($AP$) for that system. Atomic propositions can be thought of as the simplest Boolean observations one can make from the state of the system. These can then be combined in more complex expressions using standard Boolean operators such as "not" ($\neg$), "and" ($\wedge$), "or" ($\vee$), and "implies" ($\Rightarrow$). To describe behaviors over time, LTL introduces additional "temporal" operators referred to as "next", denoted by $\bigcirc$ or $X$, "eventually" (or "finally"), denoted by $\Diamond$ or $F$, "always" (or "globally"), denoted by $\square$ or $G$, and "until" denoted by $\mathcal{U}$. The syntax of LTL is given by the following grammar:

$$\varphi ::= \texttt{true} \mid \texttt{false} \mid p \mid \neg\varphi \mid \varphi_1 \wedge \varphi_2 \mid \varphi_1 \vee \varphi_2 \mid \varphi_1 \Rightarrow \varphi_2 \mid \bigcirc\varphi \mid \Diamond\varphi \mid \square\varphi \mid \varphi_1 \mathcal{U} \varphi_2.$$

Baier et al. (2008) provide a complete treatment of LTL, but we provide a brief summary of the temporal operator semantics here. The first three operators $\bigcirc$, $\Diamond$, and $\square$ are unary, meaning they apply to only one sub-expression, and respectively express the requirement that the sub-expression is true at the immediate next moment in time, at any moment beyond and including the current timestep, or for all time beyond and including the current timestep. The until operator $\mathcal{U}$ is binary, meaning that it applies to two sub-expressions, and expresses the requirement that the first sub-expression is true at all times until the second sub-expression becomes true at some future time. By composing and nesting these operators applied to atomic propositions $p \in AP$, one can quickly express complex behaviors with simple LTL specifications like the example in Figure 2.

Signal Temporal Logic (STL) (Maler & Nickovic, 2004) is a popular formalism that extends LTL in order to specify behaviors for real-valued signals defined over continuous time intervals. Using STL requires that each atomic proposition $p \in AP$ is associated with a real-valued function of the system state $\mu : S \to \mathbb{R}$, and the boolean value of the proposition in a state $s$ is determined by the value of $\mu(s) \geq c$ for some threshold value $c \in \mathbb{R}$. Furthermore, the temporal operators are extended to consider the value of sub-expressions over a specific future time interval $[a, b]$ with $a \in \mathbb{R}, b \in \mathbb{R} \cup \infty$. The resulting grammar for STL expressions is

$$\psi ::= \mu(s) \geq c \mid \neg\psi \mid \psi_1 \wedge \psi_2 \mid \psi_1 \vee \psi_2 \mid \psi_1 \Rightarrow \psi_2 \mid \Diamond_{[a,b]}\psi \mid \square_{[a,b]}\psi \mid \psi_1 \mathcal{U}_{[a,b]}\psi_2,$$

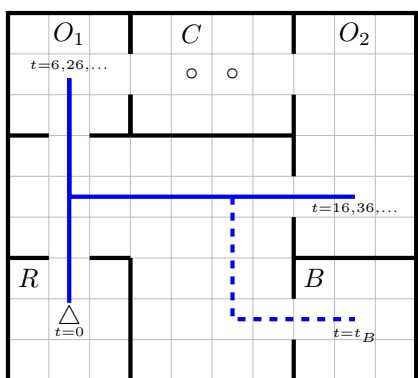

(a) The office building grid-world environment with two office rooms ($O_1$ and $O_2$), a conference room ($C$), a storage room ($R$), and a break-room ($B$).

$$\underbrace{\Box\Diamond O_1 \wedge \Box\Diamond O_2}_{\text{Recurrence}} \wedge \underbrace{\Box C_{\text{busy}} \Rightarrow \neg C}_{\text{Safety}} \wedge \underbrace{\Diamond B}_{\text{Guarantee}}$$

(b) An LTL expression describing a language (i.e., a set of behaviors) that requires the two offices are visited infinitely often, the conference room is never visited when it is occupied, and the break-room is eventually visited at least once. According to the standard classification in Manna & Pnueli (1990), these are recurrence, safety, and guarantee sub-expressions.

$$t = \quad 0 \;\ldots\; 6 \;\ldots\; 16 \;\ldots\; 26 \;\ldots\; 36 \quad\cdots\quad t_B \quad\cdots$$

$$L(s_t) = \left\{\begin{matrix}R\\C_{\text{busy}}\end{matrix}\right\} \left\{\begin{matrix}O_1\\C_{\text{busy}}\end{matrix}\right\} \left\{\begin{matrix}O_2\\C_{\text{busy}}\end{matrix}\right\} \left\{\begin{matrix}O_1\\C_{\text{busy}}\end{matrix}\right\} \left\{\begin{matrix}O_2\\C_{\text{busy}}\end{matrix}\right\} \cdots \left\{\begin{matrix}B\\C_{\text{busy}}\end{matrix}\right\} \cdots$$

(c) The labeling for each state in the example behavior, through which one can determine if the behavior satisfies the LTL expression.

Figure 2: A typical discrete domain modeling a single agent in a building with several rooms. The model is defined with atomic propositions corresponding to the agent's presence in each room and the occupancy status of the conference room ($AP = \{R, O_1, O_2, C, B, C_{\text{busy}}\}$ using the symbols in 2a). The example behavior in blue depicts the agent exiting the starting room, cycling infinitely between visiting the offices, and at some point visiting the break-room. An example LTL expression (2b) can be used to specify complex behaviors and is evaluated over sequences of labels (2c) obtained via the labeling function $L : S \to 2^{AP}$.

with timing parameters $a, b$ where $b > a$ and threshold parameter $c$. Expressions in STL, in addition to supporting the true-or-false semantics of LTL expressions, support a quantitative semantics defined by a "robustness" function $\rho$ that measures how much a sampled signal satisfies or violates the property specified by an STL formula $\psi \in \Psi$. We refer the reader to works by Maler & Nickovic (2004) and Dawson & Fan (2022) for the original definition and a more recent summary of STL semantics.

**Automata** A class of abstract machines termed FSAs provide a very useful alternative formalism for specifying behaviors. There are two broad classes of automata meant to represent languages with finite-length elements (i.e., sets containing only finite behaviors) and $\omega$-automata, which are capable of representing languages with infinite elements. More specifically, these are called regular and $\omega$-regular languages. The former class of automata has a single, well-established representation, so we elaborate here on the more complex $\omega$-automata.

An $\omega$-automaton is, like LTL, used to represent sets of sequences composed of elements in the set $2^{AP}$. However, we note that $\omega$-automata are strictly more expressive than LTL and that every LTL expression can be translated to an $\omega$-automaton (Gastin & Oddoux, 2001). An automaton is able to represent such sets (i.e., behaviors) by defining a procedure for reading one of these sequences element-by-element through which the sequence is determined to be a member of the intended set or "accepted". An example automaton for an LTL expression from the domain in Figure 2 is shown in Figure 3. This automaton is always in some state $q \in \{q_0, q_1, q_2\}$, and starts in the state $q_0$. As states are encountered in the system, the labeling function yields corresponding labels $L(s) \in 2^{AP}$ that then induce some change in the automaton's internal state as determined by the automaton's edges. The resulting sequence of internal automaton states then, based on some acceptance condition, determines the membership of the input sequence in the set. There are multiple different kinds of $\omega$-automata that differ in their acceptance condition (Hahn et al., 2022; Baier et al., 2008), of which the Rabin and Büchi acceptance conditions are the most prevalent.

**Formal Synthesis and Verification** Formal synthesis is the process of generating correct-by-construction programs that satisfy semantic correctness requirements expressed by FSs (Solar-Lezama, 2023). Classically, these methods have been applied to single-input, single-output programs expressed in digital circuits (Church, 1963), functional programs (Manna & Waldinger, 1980), and general modern programming languages (Solar-Lezama, 2009; Alur et al., 2013; 2018; Srivastava et al., 2010; Feser et al., 2015). Reactive synthesis, developed for applications to concurrent programs, instead focuses on programs that continuously receive inputs from their environment and produce corresponding outputs (Pnueli & Rosner, 1988; Bloem et al., 2012). Naturally, these reactive synthesis techniques were adapted to produce robotic controllers due to their similar computational requirements (Kress-Gazit et al., 2009). Alternatively, formal verification seeks to prove that *existing* programs satisfy a formal specification. A central class of these techniques is model checking, which operates on a discrete model of the program. By exhaustively exploring all possible executions of the model and checking them against an automaton encoding the specification, model checking can either confirm universal satisfaction or produce a counterexample

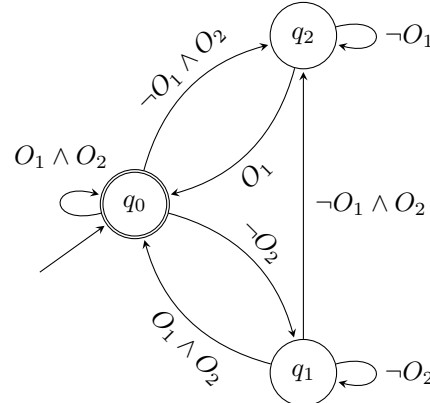

Figure 3: A Deterministic Büchi Automaton (DBA) for $\phi = \Box\Diamond O_1 \land \Box\Diamond O_2$. Intuitively, as the agent periodically encounters states $s$ satisfying $O_1 \in L(s)$ and $O_2 \in L(s)$, the automaton moves from $q_0$, to $q_1$, to $q_2$, and back to $q_0$. As a result, the accepting state $q_0 \in F$ is visited infinitely often and the sequence is deemed to satisfy $\phi$.

trace that violates the specification (Baier et al., 2008). One often performs reachability analysis as part of model checking, which focuses on determining if the system can ever reach an undesirable or target state from another state. For example, it can be used to verify that a concurrent program never enters a deadlock state. Adjacent to formal verification are other validation methods that provide weaker guarantees. Falsification attempts to find a counterexample input under which the system violates the specification. While effective at identifying bugs, falsification does not offer any guarantee of correctness in the absence of a counterexample, unlike formal verification, which aims to provide a proof of correctness across all executions.

Although there are many more specification formats and techniques within FMs, we conclude our background summary here as the majority of work seeking to bridge FMs with DL for robotic control has focused on the above fundamental methods.

# 3 Formal Methods in Robot Policy Learning

In this section, we survey the current research integrating FMs into policy learning, grouped by the major policy learning paradigm they belong to. This overall categorization and constituent subcategories are depicted in Figure 4. We also note that these categories largely align with the different communities working on robotics, each with different motivations for introducing DL or FMs to existing methods. Online methods in particular introduce DL to generalize more traditional approaches (Fainekos et al., 2005; Kress-Gazit et al., 2009; Wongpiromsarn et al., 2010; Smith et al., 2011; Ding et al., 2011; Chu (Dennis) Ding et al., 2011; Belta & Sadraddini, 2019) to settings where they are otherwise inapplicable. These include settings with unknown and complex system models, partial observability, and stochasticity, settings where standard optimization approaches are too slow, or settings where complex decisions must be made that consider longer, coarser horizons or partially satisfiable specifications. Offline methods, however, typically come from the robotics or DL for control communities and use FMs to gain support for more complex tasks and improve their safety and reliability. These research trends and their associated challenges are described in the following subsections.

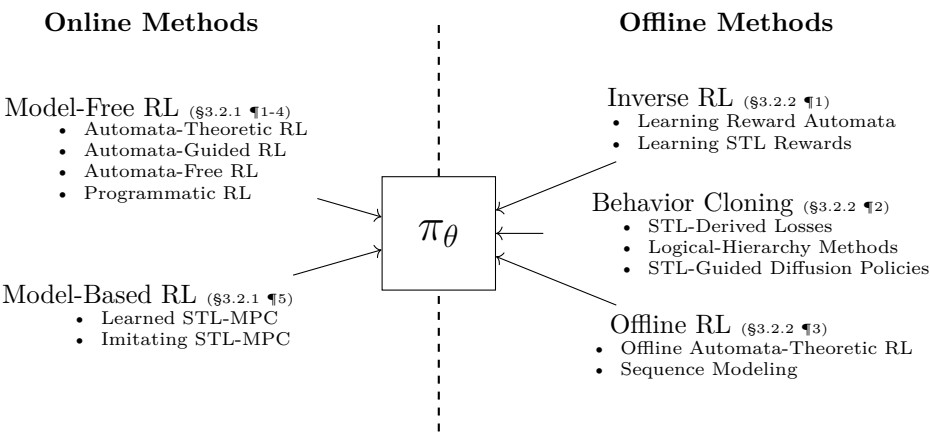

Figure 4: The research in FM-informed policy learning can be organized into categories corresponding to five major robot policy learning frameworks. The first two, model-free and model-based RL methods, are online methods that rely on environment interaction during learning. Within model-free RL, there are algorithms that learn from automata-theoretic rewards, algorithms that use automata for guidance without explicit rewards, algorithms that do not rely on automata at all, and algorithms based on programmatic policies. Model-based RL primarily extends optimal control techniques, such as Model Predictive Control (MPC), with DL components to enhance performance or versatility. The remaining three categories, Inverse Reinforcement Learning (IRL), Behavior Cloning, and offline RL, are primarily offline methods that learn from pre-collected demonstration datasets. Their constituent subfields include learning FSs as reward functions for IRL, augmenting behavior cloning losses with FSs, and training sequence models to generate actions conditioned on high FS satisfaction.

## 3.1 Online Policy Learning

**Automata-Theoretic Reinforcement Learning** The first works using DL to produce policies satisfying high-level FSs established a connection between automata-theoretic verification techniques for probabilistic systems (Courcoubetis & Yannakakis, 1995; Vardi, 1985) and model-free RL. This resulted in the field of automata-theoretic RL, where FSs are expressed as automata that monitor trajectories in a Markov Decision Process (MDP) and dispense rewards that incentivize satisfying the automaton's acceptance condition. Sadigh et al. (2014) initiated this line of research by training policies to act in product MDPs formed from a base MDP and a Deterministic Rabin Automaton (DRA). A product MDP is so named because the state space of the new MDP is formed with the Cartesian product of the plain MDP's state space and the automaton state space. This augmentation can be seen as providing a minimal "memory" necessary for obtaining Markovian rewards and policies for otherwise non-Markovian objectives. They showed that maximizing a Markovian reward function derived from the Rabin acceptance condition of a task automaton would imply maximizing the probability of satisfying the LTL formula. Although this method performed well in practice, subsequent work highlighted its limitations. Specifically, Hahn et al. (2019) demonstrated that for certain MDPs and $\omega$-regular objectives, it is not always possible to define a reward function based on the Rabin acceptance condition that yields an optimal strategy. This impossibility was later established in general (Hahn et al., 2022). In place of DRAs, DBAs emerged as an attractive alternative due to their simpler and more intuitive acceptance condition. However, simple DBAs are not ideal because they lack support for some $\omega$-regular objectives. The immediate alternative of using Non-Deterministic Büchi Automata (NDBAs), with completely unrestricted non-determinism, is also infeasible because resolving non-deterministic choices for the automaton may require an unbounded look ahead into the future (Vardi, 1985), which is impossible in a sequential-decision-making context. The solution is to rely on a special class of automata with restricted non-determinism, in which every non-deterministic choice can be resolved based solely on the history of the execution. Hasanbeig et al. (2019a) and Hahn et al. (2019) were the first to use one of these automata by defining a reward from a Limit-Deterministic Büchi Automaton (LDBA) (Sickert et al., 2016). This class of automata used in model-free RL has been further studied and formalized as "Good-for-MDPs" automata

by Hahn et al. (2020). Independently, automata-theoretic methods such as Reward Machines (Icarte et al., 2018) were developed for finite regular languages, for which a variety of reward shaping and accelerated RL algorithms have been developed (Camacho et al., 2019; Toro Icarte et al., 2022). Reward Machines have also seen greater integration with deep RL for experiments in continuous robotic domains, in contrast to prior work. Reward shaping techniques originally developed for Reward Machines in Camacho et al. (2019) have also been extended to $\omega$-automata by Bagatella et al. (2024), who achieved success in several LTL-specified robotic manipulation and locomotion tasks.

Later work in this direction relaxed assumptions about system observability implicit in the above methods. Hasanbeig et al. (2019b) and Li et al. (2024) both relaxed the assumption that one has access to the exact atomic proposition labeling function, and instead perform learning inside a probabilistically-labeled MDP. On top of the former work, Cai et al. (2021; 2024) relaxed the assumption that the input FS is always satisfiable and, in the case it isn't, learn policies that minimize task violation.

**Automata-Guided Reinforcement Learning** Instead of dispensing rewards, automata can also serve as high-level structures for guiding the execution of multiple low-level or goal-conditioned policies. For example, Jothimurugan et al. (2021) and Wang & Zhu (2025) both adopted an approach in which a separate policy for each edge of the automaton is first trained via RL. These edge-specific policies are then composed at execution time using a graph search procedure, such as Dijkstra's algorithm, to identify an optimal path through the automaton. Other approaches, such as those by Nangue Tasse et al. (2024) and Araki et al. (2021), similarly plan over automata to compose lower-level policies but instead rely on value iteration to compute an optimal composition. In recent work, Wu et al. (2025) translate LTL specifications derived from natural language specifications into Büchi automata using an equivalence voting scheme, and then apply constrained decoding techniques to ensure that planned actions adhere to paths that reach accepting states in the automata.

One can trade some generality for computational efficiency by training a single goal-conditioned policy to replace the full library of edge-specific or task-specific policies used in the above methods (Qiu et al., 2023; Manganaris et al., 2025; Guo et al., 2025). Recent works have also proposed to condition policies on real-valued embeddings of multi-subgoal sequences induced by FSs (Jackermeier & Abate, 2025), or on embeddings of entire FSs. Fundamental methods for obtaining embeddings of STL formulae have been developed by Hashimoto et al. (2022), who proposed a Word2Vec-style skip-gram model (Mikolov et al., 2013) for STL specifications, and Saveri et al. (2024), who encoded STL specifications based on the principal components of a Gram matrix defined by two datasets of random behaviors and specifications. Other techniques have also used Graph Neural Networks (GNNs) (Lamb et al., 2021; Crouse et al., 2020) for this task, using either a specification's automaton representation (Yalcinkaya et al., 2025) or the syntax tree of its temporal logic formula (Kuo et al., 2020; Vaezipoor et al., 2021) as input. These methods enable training agents to generalize to entire new tasks rather than singular goals. However, such methods sacrifice the formal soundness of their input FSs to gain from the versatility of conditional generative models. The diversity of supported tasks is also sometimes limited by assuming atomic propositions in the specifications exclusively refer to reaching goals in the environment. As a result, the most complex setting used in the evaluation of these methods has often been robotic navigation.

**Automata-Free Reinforcement Learning** Some RL methods bypass the use of automata for rewards or guidance and instead rely directly on other FS formats. Several of these are concerned with continuous-time requirements written in STL, which cannot easily use automaton-based approaches. Aksaray et al. (2016) focused on STL formulae that can be evaluated over a fixed, finite horizon length $n$ and define an MDP that augments the state space to record at all times the past $n$ states, from which a Markovian reward function can be defined. Li et al. (2017) proposed defining a non-Markovian reward based on STL robustness over entire episodes and searching for the optimal policy based on these complete policy rollouts. Other methods use similar reasoning to provide reward shaping by evaluating STL or other temporal logic formulae over the agent's history (Balakrishnan & Deshmukh, 2019; Hsu et al., 2025). Recent works have also explored directly using the gradients provided by differentiable STL specifications to train a policy online (Xiong et al., 2024; Eappen et al., 2024), although this only supports STL specifications that express requirements of the direct policy outputs (actions) as opposed to the system state. In exchange for a reduced scope of supported

FSs, these methods can be applied in many settings and can benefit from dense feedback signals. Their applications have included real-world robotic manipulation (Li et al., 2017; 2019), quadrotor goal-reaching (Xiong et al., 2024; Eappen et al., 2024), and multi-agent control tasks (Eappen et al., 2024; Hsu et al., 2025).

**Programmatic Reinforcement Learning**  There is also growing interest in RL algorithms that use *programmatic* policy representations, which more closely align with traditional program synthesis techniques. Early approaches in this direction focused on approximating RL-trained oracle policies with more interpretable, program-like representations that offer benefits such as verifiable correctness (Bastani et al., 2019) and improved generalization (Inala et al., 2020). Zhu et al. (2019) in particular took advantage of this verifiability and adopted a Counter-example Guided Inductive Synthesis (CEGIS)-style approach (Solar-Lezama et al., 2006; 2008). Their algorithm first approximates a learned policy with a symbolic program and iteratively refines it when counterexamples to its correctness are found during verification. Subsequent work sought to bypass the need for an oracle policy altogether by developing program representations that could be trained end-to-end with policy gradient methods (Qiu & Zhu, 2022). More recently, Cui et al. (2024) addressed the limitations of these approaches in long-horizon, sparse-reward settings by explicitly guiding a tree-search over possible programs with encountered rewards, thereby using programmatic policies not only for interpretable execution but also to aid in exploration. However, these methods have only been applied to environments including grid-worlds (Cui et al., 2024), two-dimensional car and quadrotor control (Inala et al., 2020), and MuJoCo locomotion tasks (Qiu & Zhu, 2022; Todorov et al., 2012), so application to more complex environments is a remaining challenge.

**Model-Based Reinforcement Learning**  Finally, research has also explored online learning of FS-satisfying policies with model-based RL. Although there are some automata-theoretic approaches in this community (Fu & Topcu, 2014), there are more that draw upon optimization-based optimal control methods (Belta & Sadraddini, 2019) and introduce learning to remove dependence on system dynamics knowledge, improve performance, or mimic decisions of experts in situations when specifications can only be partially satisfied. For instance, Cho & Oh (2018) formulated an MPC problem with constraints derived from a sequence of prioritized STL formulae and learned from a dataset of expert behavior the degree to which those constraints should be satisfied. Meng & Fan (2023) bypassed the typical sampling-based or gradient-based optimization used in MPC by directly learning a model that predicts optimal action trajectories satisfying STL specifications, enabling real-time performance. Both of these methods are primarily applied to STL-constrained autonomous driving, with additional evaluations being conducted in mobile robot and manipulator goal-reaching tasks.

Several methods simultaneously learn a model of the system dynamics along with the control policy. For example, Kapoor et al. (2020) learned a deterministic predictive model to enable MPC using an objective derived from a given STL formula. Similarly, Liu et al. (2023) learned a simple deterministic model but additionally addressed the history dependence inherent in LTL-based objectives by implementing the policy as a Recurrent Neural Network (RNN). Learning the dynamics model can also facilitate formal guarantees for the resulting policy. For instance, Cohen & Belta (2021) decomposed a high-level temporal logic requirement into a sequence of optimal control problems, each solved individually using model-based RL. The overall policy then selects among subproblem solutions based on the current state of the task automaton. The learned models and associated value functions support the construction of barrier functions (See Section 4.3) that formally certify the safety of the synthesized controller. Instead of autonomous driving, the common evaluation environment for these methods was mobile robot navigation, including a real-world deployment by Liu et al. (2023). Kapoor et al. (2020) also demonstrated goal-reaching with a manipulator robot.

## 3.2 Offline Policy Learning

Incorporating logical specifications into offline policy learning is the most recent and shallowest development of the work we survey. RL methods benefit from the fact that the logical specification can be turned into a form of online feedback in a variety of ways, but using a specification in conjunction with a set of demonstrations is more complex. However, these approaches are not burdened by the issues of sample-inefficiency found with online methods. Consequently, they more easily scale to realistic robot systems. Furthermore,

the incorporation of highly informative, compact FSs holds the promise of dramatically reducing their initial data requirements.

**Inverse Reinforcement Learning** Early methods in offline policy learning extended the product MDP construction developed for online settings and applied IRL (Ng & Russell, 2000) techniques to learn reward functions defined over the product MDP state space (Wen et al., 2017; Zhou & Li, 2018). Several of these approaches framed the problem as an instance of CEGIS, introducing the idea of generating adversarial counterexamples that highlight policy violations of the input specification in unrepresented regions of the training data, thereby improving robustness during learning (Zhou & Li, 2018; Puranic et al., 2021; Ghosh et al., 2021; Dang et al., 2023). More recent work has sought to extend IRL by incorporating Specification Mining (SM), which is the automatic generation of FSs from past experience or other informal descriptions of correctness (Ammons et al., 2002). Many techniques have been developed for this general task (Bartocci et al., 2022), of which the most significant for robotics have been learning temporal logic formulae from natural language (Liu et al., 2022) or demonstrations (Shah et al., 2018; Soroka et al., 2025) and learning Reward Machines to capture non-Markovian task rewards (Topper et al., 2022; Dohmen et al., 2022; Gaon & Brafman, 2020; Rens et al., 2020; Xu et al., 2021; Abate et al., 2023). Generally, these methods allow one to acquire learned reward functions that can be more expressive and significantly more interpretable than NN-parameterized reward functions (Liu et al., 2025b). However, we note that the policies resulting from the majority of these works have only been evaluated in very simple discrete or classic control environments, with the most recent works (Liu et al., 2025b; Soroka et al., 2025) being applied to 2D robot navigation, MuJoCo locomotion (Todorov et al., 2012), and autonomous driving.

**Behavior Cloning** Temporal logic specifications have also been leveraged in behavior cloning as additional differentiable terms in loss functions for training policies. Innes & Ramamoorthy (2020) used this approach to train dynamic motion primitive (Schaal et al., 2007) parameterized policies for pouring and goal-reaching behaviors on a real PR-2 robot. More recent approaches blend offline and online learning: an offline phase first trains a policy to mimic overall movement patterns, followed by an online phase that refines the policy by learning boundaries between discrete logical modes (Wang et al., 2022; 2024). Other behavioral cloning methods employ generative models such as diffusion models (Zhong et al., 2023b;a; Meng & Fan, 2024; Feng et al., 2024a) and flow matching models (Meng & Fan, 2025) whose inference process can be directly guided by robustness gradients from differentiable STL frameworks (Leung et al., 2023). Overall, these approaches have seen much more frequent and successful application to complex tasks including simulated maze navigation and manipulation (Meng & Fan, 2025), multi-stage real-world manipulation (Wang et al., 2022; 2024), traffic simulation (Zhong et al., 2023b;a; Meng & Fan, 2024), and real-world navigation with a Unitree Go2 quadruped robot (Feng et al., 2024a).

**Offline Reinforcement Learning** The most recent advances have focused on fully offline RL techniques (Levine et al., 2020). Due to the complexity of defining reward functions for LTL specifications, relatively few methods apply offline RL to product MDPs with the kinds of reward structures discussed in Section 3.1. One notable exception is the work by Feng et al. (2024b), who developed an approach for learning a state-option value function (Sutton et al., 1999) from a dataset with rewards based on incremental satisfaction of LTL formulae. They demonstrated superior performance to pure behavior cloning methods (Feng et al., 2024a) in 2D-Maze navigation tasks, Push-T tasks (Chi et al., 2025), and indoor navigation with a real quadruped robot. Guo et al. (2024) instead proposed an STL-controlled decision transformer that conditioned on the trajectory's robustness at each step, and they showed its successful application in several simulated robot navigation environments. Overall, this direction of offline policy learning remains underexplored and presents significant opportunities to build upon the insights developed in the online setting.

## 4 Formal Methods in Robot Policy Verification

We conclude our survey by examining research aimed at the formal verification of learned robot policies with respect to given FSs. Compared to the methods discussed in Section 3, these approaches place significantly greater emphasis on establishing formal guarantees of correct behavior, as opposed to just encouraging policies to achieve requirements given by an FS. For clarity of discussion, we uniformly frame these techniques

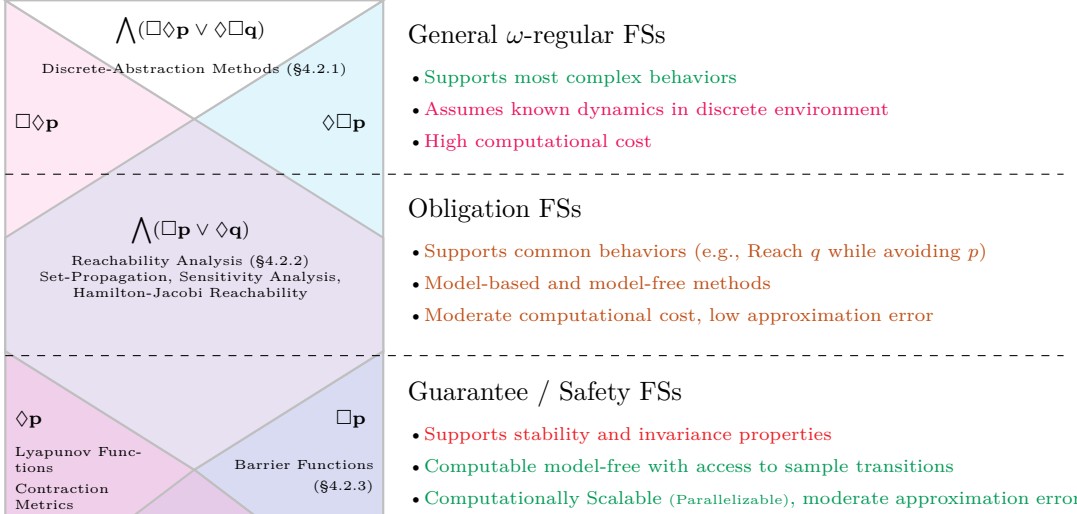

Figure 5: Our survey of robot policy verification methods is organized by three major categories of temporal FSs, visualized here using the temporal property hierarchy of Manna & Pnueli (1990). The first category (top), primarily represented by methods using discrete system abstractions, is defined by support for general $\omega$-regular FSs. The second category (middle) supports the more limited "obligation" class of FSs and is primarily represented by reachability analysis methods. The final category (bottom) supports particular forms of safety (invariance) and guarantee (eventual stability) FSs and is represented by certificate function methods. The typical capabilities of these methods are included and approximately rated as either poor in red, moderate in orange, or good in green.

as all aiming to prove that, for all initial states in some set $I$, a given NN controller $\pi$ operating within some environment will generate trajectories $\tau$ that satisfy a specification $\phi$. Under this unified view, the surveyed approaches can be compared along several axes—for example, the assumptions made about the policy, the environment, computational scalability, or the specification. We choose to primarily organize our survey around the last axis, as it naturally aligns with the others: supporting more complex specifications typically requires trade-offs in scalability and stronger assumptions on the environment model and policy. Figure 5 summarizes the surveyed work across the three major categories along this axis of FS support. While this categorization provides a useful structure, many methods within each category blur these boundaries, and we attempt to describe these liminal methods in the most appropriate location. There are also two categories that fall outside of this structure but are still important to improving the safety of robot policies: run-time monitoring and falsification methods. We also describe these and the additional capabilities they provide in the last subsections.

## 4.1 Discrete-Abstraction Methods

Discrete environment abstractions are necessary for obtaining a model under which classical verification methods are tractable, so some early approaches for verifying NN-controllers also relied on such abstractions. A prominent example is the work by Sun et al. (2019), who proposed a method that, under assumptions of linear system dynamics and a Rectified Linear Unit (ReLU)-based NN architecture, utilized a discrete system abstraction to verify closed-loop properties of a simulated, NN-controlled 2D quadrotor equipped with a LiDAR perception system. Their approach obtained these guarantees using Satisfiability Modulo Convex Programming (SMC) queries (Shoukry et al., 2017) and their custom system model. A similar strategy, with comparable assumptions but additional support for system stochasticity, was presented by Sun et al. (2022c).

While these abstraction-based methods offer considerable flexibility in the temporal specifications they can handle and provide strong correctness guarantees, the assumption that the environment can be faithfully modeled with a discrete abstraction is burdensome and unrealistic in most cases. Moreover, the discrete

abstractions themselves are prone to combinatorial explosions in complexity, which further limits their practicality (Dimitrova & Majumdar, 2014; Lindemann & Dimarogonas, 2019). As a result, these approaches have attracted less attention compared to methods that offer greater computational scalability.

### 4.2 Reachability-Based Methods

One class of computationally scalable verification methods is based on reachability analysis. While exhaustively checking all possible system behaviors is infeasible for infinite-state systems, reachability analysis provides a method to verify a more restricted set of behaviors in such continuous and hybrid systems (Bertsekas & Rhodes, 1971; Alur et al., 1995). Specifically, reachability analysis is concerned with computing approximate sets of reachable states. This reachability computation can either be performed forward in time from some initial set of states—resulting in a forward reachable set—or backward in time from some target set of states—resulting in a backward reachable set. A visualization for these two types of reachability computation is given in Figure 6. These computations then can be used to verify safety and liveness properties, often given as "Reach-Avoid specifications" (e.g., eventually reach region $A$ and always avoid region $B$). For example, if one wishes to prove that a system will always avoid an unsafe set $U$, one can either show that the forward reachable set does not intersect with $U$ or that the backward reachable set from $U$ does not include the current system state. Therefore, the forward reachable set is primarily useful for verification of the overall system when considering a specific "worst-case" initial set to start from, whereas a backward reachable set lets the agent know in general from what states safety is guaranteed and from which states it is not (Mitchell, 2007). Due to its scalability and despite some trade-offs in generality, a substantial portion of recent work has focused on reachability analysis as a viable means of verifying learned policies.

**Set-Propagation** Set-propagation methods for systems with NN controllers naturally emerged by combining existing techniques developed independently for NNs and for dynamical systems without learned components. Several early works took this integrative approach and adapted methods from both communities to analyze closed-loop systems. Early works by Xiang et al. (2018); Xiang & Johnson (2018) explicitly computed the bounds of a hyper-rectangle that over-approximated the output set of a ReLU-based NN. These bounds could then be passed to external tools for reachability analysis of systems modeled by piecewise-linear or Ordinary Differential Equation (ODE)-based dynamics. The Verisig method proposed by Ivanov et al. (2019) similarly repurposed existing system-level reachability tools by translating sigmoid-based NNs into hybrid systems modeled with ODEs, leveraging the fact that the sigmoid activation function satisfies a quadratic differential equation. This translation enabled the composition of the neural network with the environment model, allowing the verification of the resulting closed-loop system using tools such as Flow* (Chen et al., 2013). However, these early approaches frequently suffered from significant over-approximation errors during reachability analysis. A major factor contributing to this issue was the reliance on hyper-rectangular set representations, which poorly

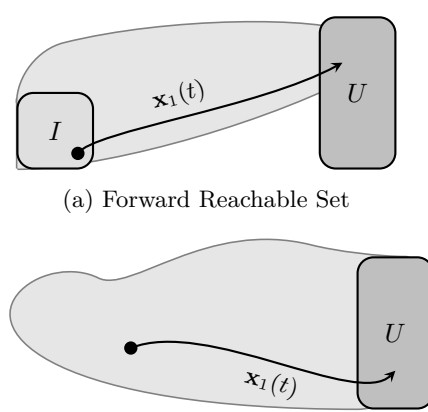

(a) Forward Reachable Set

(b) Backward Reachable Set

Figure 6: Reachability analysis for continuous and hybrid systems can verify all possible ways a system may evolve from a set of initial states $I$, or all possible ways a system might possibly evolve toward a set of unsafe states $U$.

capture the often non-convex and non-contiguous output sets produced by neural networks. More generally, directly composing the reachability computations of the NN and the dynamical system often led to rapidly compounding approximation errors due to the well-known wrapping effect (Neumaier, 1993). To address these issues, Dutta et al. (2019) proposed approximating the NN function itself, rather than just its output set, using a Taylor polynomial over a specified input range. Their tool, named Sherlock, used this approximation to create a more accurate composite model with the environment, leading to improved precision in the computed reachable set. This Taylor-model approach was later adopted and refined in follow-up versions of Verisig, which introduced techniques such as Taylor-model preconditioning and shrink-wrapping (Ivanov

et al., 2020; 2021). A related line of work replaced Taylor models with Bernstein polynomials to abstract the NN, as was implemented in the ReachNN tool (Huang et al., 2019). This approach reduced dependence on specific activation functions and supported networks with heterogeneous activations.

Several long-maintained software packages for general reachability analysis, such as CORA (Althoff, 2015) and JuliaReach (Schilling et al., 2022), have also been extended over time with implementations of reachability algorithms specifically for NNs. Newer packages such as NNV (Tran et al., 2020b) have also been created with an explicit focus on NN verification. NNV implements a wide range of NN reachability algorithms based on different set representations, including polytopes (Tran et al., 2019b), star sets (Tran et al., 2019a), zonotopes (using CORA) (Singh et al., 2018; Althoff, 2015), and ImageStar sets (Tran et al., 2020a). When combined with system-level reachability components, NNV supports verification of both feedforward and convolutional NNs, across a variety of activation functions. The follow-up work, NNV 2.0, significantly enhanced the tool's scalability and numerical accuracy, while extending support to neural ODEs, semantic segmentation networks, and RNNs (Lopez et al., 2023).

The recent work by Hashemi et al. (2023) extended set-propagation methods beyond one assumption common in the work discussed so far: the restriction to handling reach-avoid specifications. Instead, they targeted more general specifications written in STL by constructing an NN that maps initial system states to the robustness of the resulting trajectories with respect to the STL formula. Using this network, they applied standard neural network verification techniques to determine whether any initial states lead to negative robustness values. This enabled the verification of policies against a broader class of temporal specifications using only reachability-based methods.

**Sensitivity Analysis**    Despite extensive development of set propagation methods, they are fundamentally limited by their reliance on specific formats of highly accurate and deterministic system models. For the most part, they have only been successfully demonstrated in small, simulated robotic domains featuring double-integrator dynamics, quadrotors, or simplified autonomous cars. A slightly more flexible approach is available through simulating many trajectories and analyzing their sensitivity to their starting conditions or properties of the system related to trajectory divergence (Girard & Pappas, 2006; Donzé & Maler, 2007; Ramdani et al., 2008; Duggirala et al., 2013). One use case for sensitivity analysis is significantly refining the reachable sets computed by set propagation methods (Ladner & Althoff, 2023). It can also be used on its own, as in one algorithm presented in Hashemi et al. (2023) that supports a more general class of models not necessarily using ReLU activations. Their method involves dense sampling of the initial state space and estimation of the Lipschitz constant of the system's robustness function, which is then used to certify the remaining, non-sampled points. In a similar spirit, Salamati et al. (2020) propose a Bayesian inference-based technique that operates with a partially unknown model and a dataset of trajectories, constructing a posterior distribution to characterize the satisfaction probability of a given STL specification.

**Hamilton-Jacobi Reachability Analysis**    Finally, a significant body of recent research on verifying NN policies is based on Hamilton-Jacobi (HJ) reachability analysis. At its core, HJ reachability models the interaction between a control system and an adversarial disturbance as a two-player zero-sum differential game (Tomlin et al., 2000). The goal is to determine the set of states from which the system can be guaranteed to reach (or avoid) a target set despite worst-case disturbances. This problem can be formulated as a HJ Partial Differential Equation (PDE), whose solution defines a value function over the state space. The zero-superlevel set of this value function then represents the set of states from which one player is able to win, which yields the backward reachable set from a given target region (Mitchell et al., 2005).

Unlike set-propagation or sensitivity-based methods, traditional HJ reachability solves a PDE using grid-based dynamic programming, whose computational cost scales exponentially with the dimension of the state space. Addressing this scalability issue has been a central focus in research efforts aiming to apply HJ reachability to the verification of realistic robot policies. One promising direction is the use of deep-RL-style approximate dynamic programming techniques, with a contractive operator derived from the reachability PDE, to obtain a NN-approximated reachability value function (Fisac et al., 2019; Hsu et al., 2021). These methods improve scalability and, crucially, eliminate the need for an explicit model of the system, which has allowed for their application to ensure collision avoidance in more complex robotic navigation domains (Yu

et al., 2022; Manganaris et al., 2025). However, they typically require large amounts of simulated experience and may carry safety risks during training, limiting their applicability in real-world deployments unless reliable simulators or offline data are available. A second major approach, introduced by Bansal & Tomlin (2021), draws inspiration from physics-informed NNs. The HJ reachability PDE is encoded directly into a loss function that provides guidance for training a sinusoidal-NN (Sitzmann et al., 2020) to approximate the PDE solution. While this method therefore assumes known system dynamics and boundary conditions, it benefits by being significantly more data-efficient than RL-like, sample-based approaches.

Extensions of HJ reachability methods have also been proposed to support several new features. Bansal et al. (2020) developed a method for the computation of probabilistic reachable sets and applied them to a real-world system for safe navigation around humans, where reasoning about uncertainty and intent is essential. Yu et al. (2022) integrated reachability-based safety methods with RL frameworks to jointly learn safe policies and their associated safety sets. Some work has even bridged HJ reachability with STL (Chen et al., 2020) to support richer classes of safety specifications, similar to some efforts for extending set-propagation methods.

### 4.3 Certificate-Function Based Methods

A separate category of verification methods derives proofs of FS satisfaction through *certificate functions*. Certificate function methods rely on the construction of scalar functions, namely Lyapunov functions, barrier functions, and contraction metrics, whose existence imply desirable control properties. The existence of a Lyapunov function $V$ certifies that a system will eventually be driven toward a stable equilibrium region since it guarantees that $V(x)$ always decreases along system trajectories to eventually reach zero. In contrast, a barrier function $B$ certifies safety rather than goal-reaching: if the system starts within the zero sublevel set $\{x : B(x) < 0\}$, then the system remains in this safe set indefinitely, never crossing into the superlevel set $\{x : B(x) > 0\}$. Finally, contraction metrics generalize Lyapunov functions to time-varying or trajectory-tracking contexts. Instead of certifying convergence to a single point, they prove that a system can be controlled to converge exponentially toward any given reference trajectory, provided it is dynamically feasible (i.e., it is physically realizable by the system). An informative introduction and survey of these techniques is provided by Dawson et al. (2023), so we do not include a full theoretical treatment here. Instead, we focus on how certificate function methods have been developed in the context of robot policy learning and how they compare to other available verification approaches, particularly in terms of their flexibility, scalability, and applicability to learned or black-box systems.

First, the FSs verified by certificate functions are typically limited to properties concerning system invariance (e.g., remaining on one side of a barrier) or stability (e.g., eventually converging to an equilibrium point or trajectory). Therefore, such properties are not often expressed as LTL specifications, but we include them as such in Figure 5 to facilitate comparison with other verification methods. In exchange for their restricted expressiveness, certificate-based methods offer the advantages of not requiring an explicit system model and achieving favorable computational scalability.

Certificate functions are typically obtained via search over a set of differentiable functions, with constraints determined by the particular certificate type (Dawson et al., 2023). Naturally, this function space can be represented using a family of parameterized NNs, enabling the search to be cast as an unconstrained training problem by incorporating the constraints as penalty terms. The objective is then evaluated empirically over a finite set of system trajectories. This forms the basic approach underlying most certificate-learning methods (Richards et al., 2018; Srinivasan et al., 2020; Zhao et al., 2020). Subsequent research has addressed both challenges and downstream applications of this approach, including joint learning of certificates and dynamics models (Kolter & Manek, 2019), rapid adaptation of nominal certificates to new systems (Taylor et al., 2019), and new NN architectures tailored for certificate functions (Gaby et al., 2022). One particularly important challenge is that learned certificate functions resulting from this procedure are not guaranteed to satisfy their required properties. This can arise for two reasons: the NN search space may be too limited to contain a valid certificate or the empirical loss may fail to enforce constraints across the entire state space. Therefore, new synthesis techniques for certificate functions, based on CEGIS, have been introduced which adaptively refine the NN search space (Peruffo et al., 2021) and verify certificate constraints using Satisfiability Modulo Theories (SMT)-solvers (Abate et al., 2021b;a; Edwards et al., 2024).

Certificate-function-based verification also differs from other methods in that it naturally extends to obtaining formally verified controllers. Just as Lyapunov and barrier functions certify the stability and invariance of a closed-loop control system, Control Lyapunov Functions (CLFs) and Control Barrier Functions (CBFs) certify the existence of a controller for an open-loop system that can yield a closed-loop system with the desired properties. This has motivated several lines of research into certificate-regularized imitation learning (Cosner et al., 2022) and reinforcement learning (Perkins & Barto, 2003; Chow et al., 2018; Chang et al., 2019; Xiong et al., 2022; Marvi & Kiumarsi, 2021; Emam et al., 2025; Ma et al., 2021; Ahmad et al., 2025). In this context, other work has also sought to generalize CLFs and CBFs to support a broader range of STL-specified properties, through constructions such as time-varying CBFs (Lindemann & Dimarogonas, 2019). For a more comprehensive overview of certificate-based policy learning methods and of certificate functions more broadly, we again refer the reader to Dawson et al. (2023).

### 4.4 Run-Time Monitoring Methods

Another class of methods that achieve considerable scalability in exchange for weaker guarantees is run-time monitoring. As opposed to the above methods, which are capable of up-front policy verification before they are deployed, these methods continuously check during execution whether a policy is encountering an error. Once detected, the policy can be halted to allow for user intervention. For run-time monitoring with robot policies, errors can be defined in several ways that can be grouped into two categories: those defined by explicit logical specifications and those defined by heuristics or implicit specifications.

**Run-time Monitoring with Explicit Specifications**  Run-time monitoring emerged as a lightweight alternative to full model checking for complex systems, motivated in part by the combinatorial explosion of model state spaces, and is therefore classically intended to verify formally specified properties. Simulating or recording the behavior of robotic systems and evaluating specifications (e.g., STL formulae) over recorded behavior similarly allows for qualitatively evaluating otherwise intractable systems. Consequently, several foundational algorithms have been developed for efficiently evaluating the satisfaction and robustness of temporal logic specifications over system recordings, either online or offline (Donzé et al., 2013; Bartocci et al., 2018). More recent work extends these foundations to more expressive temporal logics (Bakhirkin & Basset, 2019; Bonnah & Hoque, 2022; Chalupa & Henzinger, 2023) as well as preemptive detection of errors. As opposed to purely retrospective monitoring algorithms that detect specification violation only with measurements preceding the current timestep, predictive monitoring algorithms also forecast future system behavior during monitoring to detect imminent violations using provided (Pinisetty et al., 2017) or learned predictive models (Ferrando & Delzanno, 2023; Lindemann et al., 2023; Henzinger et al., 2025). While these algorithms are generally applicable to many systems and specifications, they are, as a result, less specialized for errors encountered with robotic policies.

**Run-time Monitoring with Implicit Specifications**  Certain errors for robotic systems can be more readily defined without typical FS formats. Instead, they can be heuristically defined by sets of states labeled with known, domain-specific failure modes (Farid et al., 2022; Daftry et al., 2016; Rabiee & Biswas, 2019; Inceoglu et al., 2021; Yu et al., 2025), by anomalous behavior associated with visiting out-of-distribution states (Marco et al., 2023; Xu et al., 2024), or by feedback from large vision-language models (Agia et al., 2024). Like specification-based run-time monitors, these methods can also benefit from learned dynamics models for explicitly computing representations of future states encountered by a policy and subsequently classifying those states as invalid (Liu et al., 2024; Marco et al., 2023). The scalability of these methods has translated into their successful deployment across a variety of complex robotics domains, including real-world single-arm and dual-arm manipulation tasks (Agia et al., 2024; Liu et al., 2024; Xu et al., 2024). However, supporting more general and precise notions of correctness, as in specification-based monitoring, on complex robotic systems is an important objective for future research.

### 4.5 Falsification

Finally, several notable methods have instead focused on the problem of falsification, which is the process of showing that there exists an initial state from which the system follows a trajectory $\tau$ that does *not*

satisfy a specification $\phi$. Although this is logically separated from the normal verification problem by only a single negation operator, this form of the problem can be solved more easily and can accommodate fewer assumptions on the system, controller, and specification. One benefit of this formulation is that one can find a solution with a non-exhaustive search through the set $I$. Broad approaches to falsification can involve some form of stochastic optimization (Das et al., 2021) or robustness-guided RL (Yamagata et al., 2021). More recent approaches have enhanced the falsification process by incorporating notions of NN "coverage," aiming to ensure that the network's activation space is thoroughly explored during falsification (Zhang et al., 2023). This idea draws inspiration from prior work on automated NN testing (Sun et al., 2018; Pei et al., 2019). Other studies such as Dreossi et al. (2019) address the falsification of autonomous driving systems that combine DL for perception with classical control. Their approach alternates between searching for specification violations in the classical control component, using inputs from various abstractions of the DL perception component, and identifying errors in the DL component itself, which then inform and refine those abstractions. These methods, although not providing the same guarantees as strict verification methods, reflect the trade-off between guarantees and practicality. They still provide insight into the safety of a learned robot policy but scale better to real-world scenarios and require significantly fewer assumptions on the network architecture and model design.

## 5 Future Research Directions

In this section, we summarize the most significant remaining questions in formally specifying robot behavior, learning policies that obey those specifications, and verifying that existing learned policies do the same. For each specific area of FMs covered in our survey, we describe nascent and hypothetical research directions that we believe are promising. Moreover, we motivate these new directions with links to the larger themes we have identified as already having significance in the work surveyed above. For instance, one theme is the struggle to optimally balance formality (i.e., the strength of guarantees) and versatility (i.e., the supported specification, policy, and system complexity). Another is the increasing reliance on scalable numerical methods and decreasing favor for discrete-abstraction-based methods for synthesis and verification. We hope that these directions and the underlying trends they belong to will promote future research toward the complete resolution of safety, interpretability, and trustworthiness concerns in the modern wave of robot learning research.

### 5.1 Specifications

**Specification Mining for Robotic Systems** In robotics and artificial intelligence, comprehensive, correct, and formal specifications become harder to define as the ability to effectively model the system decreases, which is the case for the complex, real-time, continuous, stochastic, and partially-observable dynamical systems in robotics. Although SM has been explored in the context of robotics (Liu et al., 2022; Shah et al., 2018; Soroka et al., 2025; Topper et al., 2022; Dohmen et al., 2022; Gaon & Brafman, 2020; Rens et al., 2020; Xu et al., 2021; Abate et al., 2023), there are several ways in which these works are limited. For instance, extracting specifications from the full STL grammar remains intractable due to the immense search space and the limited guidance provided by datasets of observed behaviors alone. A key direction for making this problem more tractable is to incorporate external domain knowledge to constrain and guide the search. This presents a natural opportunity to leverage foundation models, which have already been shown to be highly effective complements to genetic algorithms (Novikov et al., 2025). Incorporating knowledge from a system model—either by simulating additional trajectories to enrich the dataset or by analytically reasoning about the model's behavior to rule out irrelevant specifications—is another underexplored and promising direction. Current SM methods typically focus on extracting a narrow class of specifications from specific types of datasets, most often learning unconditional formulae from complete system traces. Future work may expand this scope by targeting conditional, input-output specifications, especially from partial or noisy observations. This capability is particularly critical when learning specifications that express infinite-horizon objectives, which cannot be directly observed from finite traces (Bartocci et al., 2022).

**More Expressive Specifications** Certain robotics tasks will also require fundamentally more expressive logical languages for robotics applications, providing mechanisms to express high-level, geometrically-rich,

or abstract concepts that are difficult to represent formally. Several works have already pursued this goal by developing new temporal logics both inside and outside of robotics (Sadigh & Kapoor, 2016; Dokhanchi et al., 2019; Hekmatnejad, 2021; Balakrishnan et al., 2021; Kapoor et al., 2025), but most of these techniques have not yet been applied to robot policy learning or verification. For this last application, we believe new specification formats hold significant promise in their ability to represent interpretable, temporally structured representations of goals for artificially intelligent agents.

**Partial Observability and Stochasticity**   Obtaining and representing specifications for systems featuring stochasticity and partial observability still faces significant challenges. There are a small number of works that investigate stochasticity (Yoo & Belta, 2015; Sadigh & Kapoor, 2016) or partial observability (Kapoor et al., 2025) individually, but no methods pursuing them simultaneously. Future work may extend the approach taken by Kapoor et al. (2025) and incorporate more sophisticated probabilistic models to build formal specifications whose semantics would seamlessly incorporate multi-modal distributions for atomic propositions. This would continue the trend of increasing FS flexibility at the expense of formal specificity, so additional work may be necessary to ensure adequate guarantees are obtained when applying learning and verification methods with these FSs. Furthermore, future work can extend FS representations and implementation frameworks to accommodate these more advanced features. Differentiable STL frameworks can naturally be extended to support differentiable probabilistic models. Automaton-based representations supporting probabilistic propositions also exist (Rheinboldt & Paz, 2014) but may need adaptation for application to automata-theoretic RL.

## 5.2   Learning

**Handling more Complex Specifications**   A broad limitation in current research on learning with FSs lies in the temporal complexity of the supported FSs. We believe this gap presents an opportunity for both theoretical and practical advances. On the theoretical side, several fundamental obstacles have been identified in satisfying complex LTL specifications with RL. Addressing these challenges remains an open and compelling direction for future work. Notably, Yang et al. (2022a) showed that RL for almost all LTL objectives under their default semantics is intractable. That is, no algorithm can guarantee near-optimal performance for anything beyond the simplest LTL objectives given any finite number of environment interactions. Despite this, the work surveyed in Section 3.1 demonstrated progress by relying on reward structures derived from relaxed LTL semantics. Nonetheless, developing a unified and theoretically grounded solution to this intractability will be essential for establishing a more robust foundation for combining LTL and RL. One promising direction comes from Alur et al. (2023), who propose a discounted LTL semantics that avoids the intractability while still supporting expressive specifications. This framework offers a fertile starting point for future research. It is also the case that these more complex specifications quickly increase the sparsity of rewards. Less-theoretical research efforts can investigate practical strategies to deal with this sparsity, such as incorporating additional model-information with differentiable simulators (Bozkurt et al., 2025), introducing additional rewards in the experience replay buffer (Voloshin et al., 2023), or improving exploration with evolutionary RL algorithms (Zhu et al., 2021).

**Combining Formal and Informal Requirements**   A distinct avenue for future research could focus more on combining LTL constraints with standard RL objectives (Voloshin et al., 2022; Shah et al., 2025), as opposed to most existing algorithms that focus on solving tasks completely specified with an LTL expression. This is a critical direction, as many FSs are more meaningful when the agent pursues a competing objective, requiring it to make optimal decisions that balance task performance with formal constraints. For instance, complex specifications imposing recurrence and fairness constraints can have trivial solutions in the absence of another objective. A concrete example would be the specification "always periodically return to the charging station" in combination with a normal reward for collecting and organizing objects. The agent must then decide when to pursue objects and when to return to the charging station. Without the competing objective, however, the optimal behavior would be to remain permanently at the charging station. A possible starting point for this new direction is generalizing standard constrained MDP constraints (Altman, 1999) to accommodate functions of LTL reward or STL robustness.

**Better Offline Learning with FSs**  In the same vein, combining LTL constraints with objectives used in offline policy learning is also a fruitful and insufficiently explored direction. Offline policy learning is already considered important for safety critical settings due to circumventing the need for safe exploration, so their combination is natural. A possible broad strategy for this direction could apply innovations in automata-theoretic RL for handling non-Markovian objectives to the offline RL setting (Levine et al., 2020). Initially, research in this direction may examine enhancing methods for offline RL specifically for datasets with rewards and automaton-augmented states. Later work can then return to using standard datasets and incorporate additional guidance from specifications mined from the available data. The final and most ambitious goal for this line of research would be eventually generalizing the concept of learning optimal goal-reaching policies from suboptimal data to learning general specification-following policies from suboptimal data not necessarily following *any* specification.

### 5.3  Verification

The final major research direction we identify from the surveyed literature is in continuing to scale verification methods to support modern robot policies deployed in the real world. While recent work has already made substantial progress toward verifying satisfaction of FSs under increasingly relaxed assumptions and for more complex, high-dimensional systems, many of these methods still face significant challenges. Among the most scalable and promising lines of research are sampling-based (Lew & Pavone, 2021), dynamic-programming-based (Hsu et al., 2021; Fisac et al., 2019), and general neural-network-approximation-based (Bansal & Tomlin, 2021; Edwards et al., 2024) verification methods. We anticipate these will provide a foundation for research into more practical verification pipelines, and that this research will proceed by carefully conceding the currently offered strong guarantees in exchange for computational scalability, fewer assumptions on the environment, and support for significantly more complex models.

**Relaxing Guarantees for Scalability**  Several recent works already exemplify this trend through probabilistic extensions of reachability analysis and certificate-based methods, which allow for reasoning over stochastic system models (Lew & Pavone, 2021; Bansal et al., 2020; Castañeda et al., 2024). This probabilistic framing is particularly suitable for unstructured real-world deployment, where full certainty of future events is almost never achievable and models ought to be learned or refined online. Complex stochastic world models will also integrate well with runtime verification methods (Liu et al., 2024). While these methods may only provide probabilistic guarantees for short durations, they are the most feasible approach for long-horizon tasks in realistic, uncertain environments. This integration may also open avenues for applying these runtime verification methods to standard FSs.

**Large Policy Models**  A significant amount of future work can also be devoted to scalability concerns for more modern, complex policy models, which remains a bottleneck for many formal verification pipelines. Many current verification tools operate under restrictive assumptions about policy architectures; they often only target shallow or moderately deep feedforward networks with piecewise-linear activations. However, state-of-the-art policies increasingly rely on much more expressive and compositional architectures such as transformers and diffusion-based generative models. These models are often opaque to current analyzers, but initial efforts toward verifying these models using reachability analysis have begun to address these challenges (Shi et al., 2020; Manzanas Lopez et al., 2022; Ladner et al., 2025). Still, future research will have to further extend these techniques or pioneer new ones to handle the complex networks used in real robot policies.

**New Formal Methods for Robot Policies**  Future work may also develop FMs that are currently underutilized for learned robot policies. For instance, repairing NNs to eliminate identified faulty behavior is an active research area, with many developed techniques for localizing and efficiently retraining the responsible network components (Sotoudeh & Thakur, 2021; Dong et al., 2021; Usman et al., 2021; Sun et al., 2022a; Yang et al., 2022b; Sohn et al., 2023; Xing et al., 2024; Majd et al., 2024; Tao & Thakur, 2025). However, such techniques are most often applied to NNs used as classifiers, generative models, or discrete-action policies, and they rely on specifications whose expressiveness is tailored to those uses. Additional development is necessary for techniques to repair policies operating within the complex dynamical systems encountered by

real robots, as well as to specifications capable of expressing behavioral requirements in those domains. FMs for hierarchical or multi-agent policies are also far less explored than those for single-agent systems. Progress in this direction could draw on compositional verification techniques that decompose and verify systems using a divide-and-conquer strategy (Cobleigh et al., 2003; Alur et al., 2005). Some recent works achieve this decomposition in multi-agent systems by leveraging contracts that express each agent's individual guarantees and assumptions about other agents ("assume-guarantee contracts") (Saoud et al., 2021; Liu et al., 2025a; Kazemi et al., 2024). Still, these approaches have not yet been applied to policy learning or verification at scales needed for practical robot deployment.

## 6  Conclusion

Throughout this survey, we have shown that FMs in robot learning have the potential to substantially increase our confidence in the safe, correct, and reliable behavior of autonomous systems. To make this case, we reviewed current research in two major categories. We first surveyed techniques for learning policies that satisfy formal definitions of correct behavior (Section 3). These include methods that incorporate automata-based rewards and other forms of FS guidance into model-free and model-based RL, techniques in IRL that produce interpretable reward functions in the form of FSs, and behavior cloning and offline RL approaches that integrate supervision or conditioning based on FSs. We then examined a complementary set of verification techniques that aim to prove that a learned policy satisfies a given specification (Section 4). The verification methods range from the most formal but least scalable methods based on formulating and checking discrete environment abstractions, to the more focused, moderately scalable reachability-analysis methods, and finally to the most specific and scalable certificate function methods. We also surveyed work in the less formal but practical categories of run-time monitoring and falsification. These summaries provide a comprehensive overview of the current research on FMs applied to learned robot policies.

Beyond summarizing the state of the art, our survey provides a coherent organizational framework that can serve both new entrants and seasoned researchers in the field. For readers newly approaching formal methods in robot learning, we have included accessible illustrations, examples, and conceptual overviews that clarify foundational ideas and map out the landscape of existing work. For experts, our survey provides a concise taxonomy of techniques, starting points for deeper inquiry into all the surveyed subfields, and connections between them that are not often discussed. Most importantly, by summarizing current trends in FM-informed policy learning and verification research, we identify a number of underexplored questions and promising future directions (Section 5). In particular, we emphasize the need to increase the expressiveness and scalability of current FMs to support learning and verification in real-world robotic systems. We believe that achieving this goal will be essential for ultimately deploying robots that are not just performant, but provably safe and trustworthy.

## Acknowledgments

This material is based upon work supported by the Air Force Office of Scientific Research under award number FA9550-24-1-0233. Any opinions, findings, and conclusions or recommendations expressed in this material are those of the author(s) and do not necessarily reflect the views of the United States Air Force.

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

## A Acronyms

**DL**             Deep Learning

**ML**             Machine Learning

| | |
|---|---|
| **FM** | Formal Method |
| **FS** | Formal Specification |
| **SM** | Specification Mining |
| **RL** | Reinforcement Learning |
| **IRL** | Inverse Reinforcement Learning |
| **LTL** | Linear-time Temporal Logic |
| **STL** | Signal Temporal Logic |
| **MDP** | Markov Decision Process |
| **FSA** | Finite State Automaton |
| **DBA** | Deterministic Büchi Automaton |
| **LDBA** | Limit-Deterministic Büchi Automaton |
| **NDBA** | Non-Deterministic Büchi Automaton |
| **DRA** | Deterministic Rabin Automaton |
| **CEGIS** | Counter-example Guided Inductive Synthesis |
| **ReLU** | Rectified Linear Unit |
| **NN** | Neural Network |
| **RNN** | Recurrent Neural Network |
| **GNN** | Graph Neural Network |
| **ODE** | Ordinary Differential Equation |
| **PDE** | Partial Differential Equation |
| **MPC** | Model Predictive Control |
| **CLF** | Control Lyapunov Function |
| **CBF** | Control Barrier Function |
| **HJ** | Hamilton-Jacobi |
| **SMC** | Satisfiability Modulo Convex Programming |
| **SMT** | Satisfiability Modulo Theories |

