# OpenReview forum: "Formal Methods in Robot Policy Learning and Verification: A Survey on Current Techniques and Future Directions"
_TMLR — Accepted by TMLR_

### Review · Reviewer_NSyM · 2025-09-05

**Summary Of Contributions:**

The paper surveys formal and semi-formal methods for specifying behavior, training deep learning models, and performing quality assurance (QA), that may be relevant for robot policy learning. The review is structured along these columns: specifications, training, and QA. Further, it provides suggestions for future work at the end.

Strengths:
 1. The paper covers many relevant applications of formal methods.
 2. The paper provides a wealth of pointers to existing work on various applications of formal methods.

Weaknesses:
 1. The survey is not targeted.
 2. The paper is poorly structured and written.
 3. The paper contains factual errors.

**Additional Comments:**

A few important formatting and more minor writing issues:
- A reference for abbreviations should not be an excuse for not introducing abbreviations properly. Abbreviations such as "HJ" should be introduced in the text.
- In figure 1, the figure and the caption use different phi symbols.
- Figure 7 contains references for some sections but not for others
- The bibliography is poorly formatted. For example, abbreviations, such as STL and SpaTeL are frequently not capitalized correctly. Also, some conferences are abbreviated, while others are not, and DOIs and URLs sometimes duplicate themselves.

There are also a number of typos in the manuscript. Most notably the last sentence on page 7 just ends abruptly.

**Audience:**

Yes

**Audience Explanation:**

Formal methods are interesting for robotics in safety-critical systems, as is motivated well in the paper.

**Broader Impact Concerns:**

As this is a survey of existing work, I have no concerns about broader impact.

**Claims And Evidence:**

No

**Claims Explanation:**

The paper claims to perform a survey of formal methods in robot policy learning. Many of the surveyed papers are not about robot policy learning but general formal methods papers. For example, section 2.2.1 contains at best three citations to papers about robotics from 34 citations on about two pages. While the other papers are about general techniques that may be useful for robot policy learning, this does not match the claim that "this is a survey of FMs as applied to learning methods for robot policies".

Overall, this is weakness 1, the survey is not targeted.

If the purpose is to also survey formal methods more generally, it is unclear how the presented approaches were selected, as there is certainly a wealth more of formal method papers that might be broadly interesting for robot policy learning. Some examples under "Requested Changes".

**Requested Changes:**

Critical Changes.

### Structure & writing needs to be improved (Weakness 2)
For example,
- The preliminaries section 2.1 is clearly also important for sections 3 and 4 but the structure makes it seem like it is only important for section 2.
- Section 3 is structured along different RL technologies. This might be helpful for a review on RL but it is not clear to me why this differentiation is insightful for robotics or formal methods.
- Paragraph "Languages" in section 2.1 is about transition systems and not formal languages.
- The paragraph just above "Temporal Logic" claims that the paper focuses primarily on LTL, while many of the papers are actually on STL.
- The text does not say that "p" and phi are in the syntax equation for LTL. In contrast, the text specifies this for STL but after LTL was introduced.
- The discussion of DRAs vs DBAs vs LDBAs is duplicated on page 8 and 12.
- The paragraph on falsification on page 14 is severely misplaced, as it does not relate to discrete systems or hybrid systems.
- The MILP-based approaches discussed on page 17 do not employ set propagation but are placed in a paragraph titled "set propagation".
... and more.

Overall, it seems like the paper did not undergo proofreading. This is also evident from formatting errors, such as the many citations that are not set in parentheses.

### Factual Errors (Weakness 3)
- The presented LTL semantics are incorrect. For example, the always operator requires phi not only for all times in the future but also the current point in time.
- The discussion of STL forgets to mention that punctual intervals with a = b are disallowed, although this is essential for the decidability of STL.
 - The discussed finite automata are more expressive than LTL. They do not recognize *the same* languages.

An adjacent point is the usage of the term "synthesis" in the paper. It correctly gives the standard definition of synthesis as generating a system that *provably* obeys a specification in section 3.1. However, in the remainder of the paper, "synthesis" is used much more liberally. The first paragraph in section 3 even equates synthesis and learning, which are two very different things from a formal methods point of view. As such, also the fundamental organization as depicted in figure 1 does not make sense from a formal methods point of view, since synthesis does not require verification afterwards. A more apt title for section 3 would be "learning with formal specifications".

### Relevant Directions not Covered
- When looking at learning and verification, repair is a key technique for obtaining a satisfactory pipeline. See, for example "Provable repair of deep neural networks" by Sotoudeh and Thakur (2021).
- There are other recent tools for verifying neural-network-controlled cyber-physical systems besides NNV, such as Cora https://tumcps.github.io/CORA/ or JuliaReach https://juliareach.github.io/ that are not discussed, while NNV is.

### Further Issues
- The reference (Fang et al., 2024) does not support the claim on page 1. Rather, Fang et al. state that NNs can learn out-of-distribution detection.
- In what regard are formal specifications "richer" than arbitrary reward functions? (4th paragraph on page 2)
- What is understood by "value misalignment"? (same paragraph)
- What is meant by "temporally extended behaviors"? (1st paragraph on page 5)
- The "Many followup works in SMs" in the 3rd paragraph on page 7 are not cited.
- What is a "plain" FSA? (1st paragraph on page 12)
- What is meant by "approximate methods"? (2nd paragraph on page 15)
- What is meant by "dynamically feasible"? (3rd paragraph on page 15)

---

Other changes that would strengthen the paper.

### Length
- It is unclear if all of the included material is vital for the paper. For example the section on automata could be removed.

### Further Directions
- Runtime monitoring is another formal method that can be relevant for robot policy learning.

---

> ### Author Response · Authors · 2025-11-19
> **Rebuttal by Authors (Part 1)**
>
> Thank you very much for your thorough review of our work. We have categorized your points according to whether they discuss the surveyed content, organization, factual points, and miscellaneous points, and we have addressed them below. We would be happy to provide any additional clarifications if needed. We sincerely appreciate your feedback.
>
> ## Content Issues
>
> > Runtime monitoring is another formal method that can be relevant for robot policy learning.
>
> Thank you for bringing this work to our attention, and we agree that this is important work for ensuring learned robot policies behave appropriately. We have added a discussion on runtime monitoring methods in Section 4.4, highlighted in blue.
>
> > There are other recent tools for verifying neural-network-controlled cyber-physical systems besides NNV, such as Cora https://tumcps.github.io/CORA/ or JuliaReach https://juliareach.github.io/ that are not discussed, while NNV is.
>
> Thank you for this point. We agree these tools are of interest to our audience, so we have modified our discussion on modern reachability tools to more broadly discuss the current options.
>
> > When looking at learning and verification, repair is a key technique for obtaining a satisfactory pipeline. See, for example "Provable repair of deep neural networks" by Sotoudeh and Thakur (2021).
>
> Thank you for bringing this work to our attention. We agree that this is an interesting formal method for application to neural networks. However, there appears to be almost no work focused on repair of neural-network policies. Therefore, we have decided to mention this line of research in the future research directions, Section 5.3, under the paragraph “New Formal Methods for Robot Policies”, highlighted in blue.
>
> ## Organization
>
> > The preliminaries section 2.1 is clearly also important for sections 3 and 4 but the structure makes it seem like it is only important for section 2.
>
> We agree that elements present in Section 2.1 might be relevant also for section 3 and 4, so we have decided to shorten and restructure this section as a general preliminaries section.
>
> > Paragraph "Languages" in section 2.1 is about transition systems and not formal languages.
>
> While we begin this paragraph to discuss discrete transition systems, we do so in order to explain how behaviors of systems can be described as sequences of states, and then how specifications for systems correspond to sets of these sequences (i.e., languages). To reduce this confusion, we have removed the label from this paragraph.
>
> > The discussion of DRAs vs DBAs vs LDBAs is duplicated on page 8 and 12.
>
> While we originally had discussed these works in both sections to separately cover the research on using automata as rewards and the learning algorithms that exploit them, we have now decided to consolidate this discussion in the Automata-Theoretic Reinforcement Learning paragraph under Section 3.1. Any changes are highlighted in blue.
>
> > Section 3 is structured along different RL technologies. This might be helpful for a review on RL but it is not clear to me why this differentiation is insightful for robotics or formal methods.
>
> We respectfully disagree with this comment. Section 3 broadly analyzes approaches used in robotics to perform end-to-end robot policy learning. Most of these studies leverage either imitation learning, which belongs to the class of supervised learning methods applied to correlated data, and, as mentioned by the reviewer, RL-based methods. As a result, we believe that our Section 3 clearly surveys the state-of-the-art to formal methods when used within the context of robot policy learning.
>
> > The paragraph on falsification on page 14 is severely misplaced, as it does not relate to discrete systems or hybrid systems.
>
> Thank you for pointing this out. We agree that this introduction of falsification could be improved and we have moved its introduction, along with the introduction of classical formal synthesis verification, to our new preliminaries section (now in Section 2, under the label “Formal Synthesis and Verification”)

---

> ### Author Response · Authors · 2025-11-19
> **Rebuttal by Authors (Part 2)**
>
> ## Factual
>
> > The always operator requires phi not only for all times in the future but also the current point in time.
> > The discussion of STL forgets to mention that punctual intervals with a = b are disallowed, although this is essential for the decidability of STL.
> > The text does not say that "p" and phi are in the syntax equation for LTL. In contrast, the text specifies this for STL but after LTL was introduced.
> > The discussed finite automata are more expressive than LTL. They do not recognize the same languages.
> >The paragraph just above "Temporal Logic" claims that the paper focuses primarily on LTL, while many of the papers are actually on STL.
>
> Thank you for bringing these to our attention. We have now revised our introduction of LTL/STL semantics to more precisely define the semantics for the temporal operators, the meaning of “p” in the LTL grammar. We have also clarified that omega-automata are strictly more expressive than LTL, and we have modified our introduction of formal specifications to more broadly refer to temporal logics rather than just LTL. These changes are highlighted in blue in Section 2.
>
> > An adjacent point is the usage of the term "synthesis" in the paper. It correctly gives the standard definition of synthesis as generating a system that provably obeys a specification in section 3.1. However, in the remainder of the paper, "synthesis" is used much more liberally. The first paragraph in section 3 even equates synthesis and learning, which are two very different things from a formal methods point of view.
>
> We agree that using “learning” to describe the works we survey is more accurate than “synthesis”, so we have changed this terminology throughout the survey. Any changes are highlighted in blue.
>
> > The reference (Fang et al., 2024) does not support the claim on page 1. Rather, Fang et al. state that NNs can learn out-of-distribution detection.
>
> We choose to cite Fang et al., 2024 for this claim to show that it is in fact impossible in many settings to learn to even identify states outside of the training distribution, which also entails being unable to learn how to act in those states. However, to provide different evidence of specifically policy learning methods struggling with generalization beyond the training distribution, we now cite Ross and Bagnell, 2010 for showing the quadratic increase in error for policies due to deviation from the expert state distribution and Park et al., 2024 for empirically showing the limitations of Offline RL due to distribution shift between the demonstrator and learned policy.
>
> ## Miscellaneous
>
> > The MILP-based approaches discussed on page 17 do not employ set propagation but are placed in a paragraph titled "set propagation".
>
> Thank you for this point. We initially placed these MILP-based approaches in this section as they are using MILP to determine the extrema of a polytope representing the output set of a neural network controlled dynamical system. This could be considered similar to determining the propagation of a set, although the entire boundary of the set is not computed. However, as these methods are mostly subsumed by the work described later in that section, we have now removed these methods from our survey.
>
> > many citations that are not set in parentheses.
>
> Thank you for bringing this to our attention. We have carefully checked all the citations and modified them accordingly.

---

> ### Author Response · Authors · 2025-11-19
> **Rebuttal By Authors (Part 3)**
>
> > In what regard are formal specifications "richer" than arbitrary reward functions? (4th paragraph on page 2)
>
> We consider formal specifications to be strictly more expressive than reward functions due to the fact they can directly express non-markovian requirements on entire trajectories produced by a policy. They also are able to naturally express requirements that ought to be only satisfied rather than maximized.
>
> > What is understood by "value misalignment"? (same paragraph)
> By “value misalignment” we mean that the deployed policy does not maximise or promote the same value-function that the user or system designer intends. In this sense our use of “value misalignment” falls under the broader umbrella of the Alignment problem in AI: how to ensure that autonomous agents behave in accordance with human intentions, preferences or values.
> > What is meant by "temporally extended behaviors"? (1st paragraph on page 5)
>
> By “temporally extended behavior”, we refer to actions or decision patterns that unfold over long time horizons and depend on information spanning multiple time steps, rather than being determined solely by the current state. Such behavior requires the agent to reason about sequences of events and maintain or infer memory of past observations that may not be explicitly encoded in the state representation.
>
> > What is a "plain" FSA? (1st paragraph on page 12)
>
> We use the modifier “plain” in this context to refer to a finite-state automaton which is not an omega-automaton. Due to this difference, the methods described in that paragraph do not work with omega regular objectives but only regular objectives.
>
> > What is meant by "approximate methods"? (2nd paragraph on page 15)
>
> By “approximate methods” in reachability analysis, we refer to techniques that compute over-approximations or under-approximations of the true reachable set of a dynamical or hybrid system, rather than exact characterizations. Exact reachability is often computationally intractable or even undecidable for nonlinear, hybrid, or neural network–controlled systems. Approximate methods trade precision to achieve tractability.
>
> > What is meant by "dynamically feasible"? (3rd paragraph on page 15)
>
> A dynamically feasible trajectory is a sequence of states through which the system could evolve while respecting the constraints of the system dynamics. For a discrete-time system, this is equivalent to saying that for all pairs of states x_i, x_i+1 in x_1, x_2, x_3, …, there exists some action u such that x_i+1 = f(x_i, u).
>
> > Length: It is unclear if all of the included material is vital for the paper.
> > Many of the surveyed papers are not about robot policy learning but general formal methods papers.
>
> Thank you for raising this issue. Although we feel that almost all of the papers we included in our survey were relevant for our audience, either as being directly related to robot policies or for defining formal specifications in domains that are also found in robotics, we agree that the length added by these methods may have been excessive. Therefore, we have reduced our discussion  of the works less directly connected to robot policies and restructured our survey accordingly.
>
> > For example the section on automata could be removed.
>
> We chose to include our introduction to omega automata in order to make our survey self-contained for the robotics researcher who may not be familiar with the concept of automata. However, we agree that some details in this section were unwarranted and we have reduced its length in our revised version.
>
> > A reference for abbreviations should not be an excuse for not introducing abbreviations properly. Abbreviations such as "HJ" should be introduced in the text.
> > In figure 1, the figure and the caption use different phi symbols.
> > Figure 7 contains references for some sections but not for others
> > There are also a number of typos in the manuscript. Most notably the last sentence on page 7 just ends abruptly.
>
> Thank you for bringing these to our attention. We have now corrected the mismatch in symbols and all typographic errors in our updated version. We wish to note that all of our abbreviations, including “HJ” are introduced in the text by use of the LaTeX acronym package. As HJ was introduced in the introduction but later more fully discussed in Section 4, we have made it so that the HJ abbreviation is introduced again in Section 4. For Figure 7, we wish to express that references are intentionally given for only the top-level subsections for each method (i.e., discrete-abstraction methods, reachability analysis, and certificate functions). The certificate function methods are given one reference to section 4.2.3 to reduce redundancy and the lower-level paragraphs are not referenced on purpose. Finally, while we respectfully disagree with the last comment, as the last sentence on page 7 was a complete sentence, we have removed this sentence in our restructuring.

---

> > ### Comment · Reviewer_NSyM · 2025-11-24
> > **Good direction but issues remain**
> >
> > Dear authors,
> > I appreciate the changes you made, and I believe you are moving in a good direction. However, several of the issues I outlined remain unaddressed, and the revision has also raised new issues. In my opinion, this paper still needs substantial work to become a valuable review article. I list the most salient points below:
> >
> > **Structure**
> > > We respectfully disagree with this comment. Section 3 broadly analyzes approaches used in robotics to perform end-to-end robot policy learning. Most of these studies leverage either imitation learning, which belongs to the class of supervised learning methods applied to correlated data, and, as mentioned by the reviewer, RL-based methods. As a result, we believe that our Section 3 clearly surveys the state-of-the-art to formal methods when used within the context of robot policy learning.
> >
> > I agree that it is *possible* to organize these works along the RL techniques applied. However, that does not mean that this is *insightful* to organize these approaches along those techniques, which was also my original point.
> >
> > **Content**
> > As described in the revision, the discussed works on runtime monitoring do not leverage formal methods in that they do not rely on formal specifications. Because of this, I consider them out of place for this review. However, there is a wealth of works on runtime monitoring using formal specifications, for example, by Ezio Bartocci or Thomas A. Henzinger.
> >
> > > Thank you for bringing this work to our attention. We agree that this is an interesting formal method for application to neural networks. However, there appears to be almost no work focused on repair of neural-network policies.
> >
> > This is not the case. The ACAS Xu and VCAS case studies that are widely used in neural network verification and repair are concerned with neural-network policies. There are further works that target other policies as well, such as Yang et al.
> > "Neural Network Repair with Reachability Analysis" in FORMATS 2022.
> >
> > **Further Issues**
> > Thank you for your explanations on the terms and formulations I asked questions about. These need to be introduced in the text as well, as they are overall non-standard terms.

---

> > > ### Author Response · Authors · 2025-11-26
> > > **Re: Good direction but issues remain**
> > >
> > > We thank the reviewer for their engagement and have responded to the newly raised points.
> > >
> > > > I agree that it is possible to organize these works along the RL techniques applied. However, that does not mean that this is insightful to organize these approaches along those techniques, which was also my original point.
> > >
> > > Thank you for this helpful clarification. We understand the reviewer's concern. However, we would like to highlight that our intention was not to categorize works by specific RL techniques, but rather by broader **policy learning paradigms** (e.g., model-free learning, model-based learning, imitation learning). We chose these paradigms because they correspond to distinct ways that the surveyed papers **integrate formal methods into the learning process**. Presenting the literature through these paradigms serves two purposes:
> > >
> > > **1. It reveals structural patterns in how formal methods influence learning.** For example, automata-theoretic RL methods extend model-free RL, and differentiable temporal-logic-based imitation learning methods extend behavior cloning. Organizing by paradigm clarifies these cross-paper patterns, showing how FMs inform different policy learning techniques.
> > >
> > > **2. It helps readers map FM-augmented methods onto the learning strategies they already understand.** Our target audience comes from diverse backgrounds in robot learning; anchoring FM-augmented methods to familiar paradigms allows them to more readily identify which FM tools are relevant to their own research workflows.
> > >
> > > We hope this clarifies the insight we aim to convey and makes the organizational choice more compelling.
> > >
> > > > As described in the revision, the discussed works on runtime monitoring do not leverage formal methods in that they do not rely on formal specifications. Because of this, I consider them out of place for this review. However, there is a wealth of works on runtime monitoring using formal specifications, for example, by Ezio Bartocci or Thomas A. Henzinger.
> > >
> > > Thank you for this helpful comment. We agree that runtime monitoring methods that are explicitly grounded in formal specifications represent the core of formal-methods-based verification, and we appreciate the reviewer pointing us to the extensive body of work in this area. Unfortunately, we find the majority of these specification-based monitoring methods are either not applicable in the same modern robotic domains considered by the rest of the survey [1] or using restricted specifications that are functionally equivalent to the ones we current discuss [2, 3]. Nevertheless, we agree that these more formal methods are important and have included references to them under subsection 4.4 to explicitly motivate them as a promising starting point for extending formal-specification-based monitoring into modern robotic learning settings.
> > >
> > > Although the monitoring methods we have already included do not use formal specifications, we believe they are still appropriate for inclusion in our survey, considering that they are still an important method for formally verifying interesting properties of robot policies.
> > >
> > > [1] Bartocci, E., et al. (2018). Specification-Based Monitoring of Cyber-Physical Systems: A Survey on Theory, Tools and Applications.
> > > [2] Henzinger, T. A., et al. (2025). Predictive Monitoring of Black-Box Dynamical Systems.
> > > [3] Yu, E., et al. (2025). Neural control and certificate repair via runtime monitoring.
> > >
> > > > The ACAS Xu and VCAS case studies are concerned with neural-network policies. There are further works that target other policies as well
> > >
> > > Thank you for clarifying this point. We concede that there are existing works that apply repair to neural-network policies and have modified subsection 5.3 accordingly, but we believe it is still justified to discuss these works in the future directions section of our survey focused on learning *robot policies*. This is primarily because these repair methods, based on their application in these case studies, can only repair discrete-action-space policies (essentially classifiers) based on ground-truth labels provided by an existing finite table. The majority of robot policies use a continuous action space and are used in domains for which a ground-truth policy cannot be obtained. As a result, using these methods for learning or verifying robot policies, in the same sense as the currently-included methods, is not immediately possible. While the work by Yang et al. is more aligned with the setting of robot policy verification, we still believe it is more appropriate to discuss the overall subfield containing this work as a starting point for future research focused on domains with the complexity of real robotic systems.
> > >
> > > > These need to be introduced in the text as well, as they are overall non-standard terms.
> > >
> > > To improve clarity, we have replaced “richer” with “more expressive”, rewrote the sentence using “misalignment”, and included a definition for “dynamically feasible” in our new revision.

---

### Review · Reviewer_T8XK · 2025-09-13

**Summary Of Contributions:**

This paper provides an overview of the integration of Formal Methods (FMs) into robot policy learning, focusing on Deep Learning (DL)-based approaches in robotics. It organizes the paper following a three-part pipeline:
- specification: (acquiring and representing formal behavioral requirements, such as using temporal logics like LTL and STL, or mining them from data),
- synthesis: (learning policies that satisfy these specifications, including FM-guided reinforcement learning, imitation learning, and inverse reinforcement learning),
-  verification: (proving or falsifying that learned policies meet specifications, via methods like reachability analysis, certificate functions, and falsification techniques).

The paper highlights representative techniques in each area, compares their scalability (e.g., handling high-dimensional systems) and expressiveness (e.g., support for stochastic or partially observable environments), and discusses how they enhance robot safety, interpretability, and trustworthiness. Additionally, it identifies key gaps (e.g., handling complex LTL specifications, partial observability, and scalability for modern architectures like transformers) and proposes promising future directions (e.g., combining LTL with offline RL, probabilistic extensions for real-world deployment, and leveraging foundational models for specification mining). Overall, it serves as a valuable resource for the research community, aiming to guide research toward provably safe and reliable robots by integrating formal methods.

pros:
- Formal methods in robot policy are an important topic that addresses the growing need for safety and trustworthiness in DL-based robotics.
- This paper provides a broad range of works related to formal methods in robot policy learning, which is useful to the community.

Cons
- Writing should be largely improved. I find it hard to follow the paper. And it took me three times of reading to figure out the major structure of the paper.
- As a survey focusing on formal methods in robot policy, this paper puts too much effort into introducing the formal specifications methods (whole section 2), for example, different formal methods such as TLT, SLT, and Automata. I strongly recommend that the authors shorten Section 2 as preliminaries or put Section 2 in the appendix to avoid hampering the reading of the whole story. Since Section 2 serves as the preliminary but not part of the robot policy.
- As two methods which are also formal methods, First Order Logic (FOL) and STRIPS planners are missing in part 2. And correspondingly, the related works are missing in both aspects, such as [1] and [2]
- I believe this is a typo on page 2, 4th paragraph, 4th line. It should be ‘’moreover , FMs allow…..’’ instead of  ‘’moreover , FSs allow…..’’


[1] Ye, Z., Arenz, O., & Kersting, K. (2025). Learning from Less: Guiding Deep Reinforcement Learning with Differentiable Symbolic Planning. arXiv preprint arXiv:2505.11661.

[2] Kienle, C., Alt, B., Arenz, O., & Peters, J. (2025). LODGE: Joint Hierarchical Task Planning and Learning of Domain Models with Grounded Execution. arXiv preprint arXiv:2505.13497.

**Additional Comments:**

N/A

**Audience:**

Yes

**Audience Explanation:**

this paper includes many related work that can be beneficial for the research community and the topic the paper addresses is an important  topic

**Claims And Evidence:**

Yes

**Claims Explanation:**

as a survey paper it cites the corresponding paper and explain each paper accordingly

**Requested Changes:**

- minor typo on page 2, 4th paragraph, 4th line should be corrected
- Section 2 should be largely improved and also shortened or maybe moved to the appendix
- The writing should be largely improved

---

> ### Author Response · Authors · 2025-11-19
> **Rebuttal by Authors**
>
> Thank you very much for your feedback. We have addressed the points you raised below, and we would be happy to provide any additional clarifications if needed. We sincerely appreciate your feedback.
>
> ## Content Issues
>
> > As two methods which are also formal methods, First Order Logic (FOL) and STRIPS planners are missing in part 2
>
> Thank you for raising this point. While planning and first-order logic are relevant topics related to robotic decision making and knowledge inference, we don’t believe these are typically considered formal methods as in the formal methods used in software systems. That is, they are not mathematically grounded techniques for specifying and verifying correctness properties of systems.
>
> ## Writing
>
> > Writing should be largely improved
> > Too much effort into introducing the formal specifications methods
> > Typos
>
> Thank you for your feedback. We have carefully reviewed and simplified the writing throughout the survey, and we have fixed all found typos. We have also taken care to simplify the introductory material in Section 2, and we have re-packaged this in a simplified preliminaries section.

---

### Review · Reviewer_ZusR · 2025-10-12

**Summary Of Contributions:**

This paper is a survey on the use of Formal Methods (FMs) in robot policy learning. The authors argue that while deep learning has enabled highly capable robot policies, these models are often fragile, difficult to interpret, and lack safety guarantees.

The survey organizes the field into three key parts:

- Specification: How to rigorously define complex, temporally-extended tasks and safety requirements for robots using formal languages like temporal logic.
- Synthesis: How to create or "learn" robot policies that are guaranteed to meet these formal specifications.
- Verification: How to formally prove that an already-learned policy correctly and safely adheres to a given specification.

The paper provides a comprehensive overview of techniques within each category, compares their scalability and expressiveness, and concludes by discussing open challenges and future research directions for developing provably safe and trustworthy robots.

**Audience:**

Yes

**Audience Explanation:**

Formal methods can potentially bridge the gap between the high performance of modern AI and the high stakes of real-world robotics. So I believe both AI and robotics researchers will be interested in this paper.

**Claims And Evidence:**

Yes

**Claims Explanation:**

As a survey paper, this paper's claims are mostly about the state of a research field, its structure, and its future trajectory. And they are generally supported by extensive citations and logical organization.

I listed a few "claims" in this paper below:

- About problem: While deep learning has led to highly performant robot policies, it has simultaneously introduced critical weaknesses. These learned policies are often "inflexible, fragile, and difficult to interpret," lack formal guarantees, and are vulnerable to issues like reward hacking and poor generalization, making them untrustworthy for safety-critical applications.


- About solution: Formal Methods (FMs), which have been indispensable for ensuring correctness in complex software systems, represent a promising approach to address the weaknesses of DL-based robot policies and transform them into "transparent, reliable, and trustworthy solutions.

- About framework: The work at the intersection of FMs and robot policy learning can be naturally and comprehensively organized into a three-stage pipeline: Specification, Synthesis, and Verification.

- About future directions: The paper identifies specific, significant gaps and promising directions for future research, such as handling more complex specifications in RL, developing better offline learning methods with FMs, and scaling verification to larger, more modern policy models.

**Requested Changes:**

Overall, I think this paper is in pretty good shape. If possible, I’d love to see a few examples of how formal methods are applied in real-world robotic systems.

---

> ### Author Response · Authors · 2025-11-19
> **Rebuttal by Authors**
>
> Thank you very much for your positive review of our work. In our updates to the survey, we have included more information on the evaluation for every method we survey, with special emphasis given to those that are deployed in real-world robotic systems. We would be happy to provide any additional clarifications if needed. Thank you again for your feedback.

---

### Decision · Action_Editor_6KWX · 2025-12-01

**Recommendation:** Accept with minor revision

**Additional Comments:**

At the end of the rebuttal, 2 reviewers recommend acceptance, whereas reviewer NSyM argues that the changes were mostly cosmetic and that the revision introduced new issues, by discussing papers on runtime monitoring that do not use formal methods. I agree that some of the issues raised by reviewer NSyM have not been fully addressed, but I think that the current submission does not require a major revision. Hence, I think that the submission can be conditionally accepted, provided that the following revisions are made:

- Formal methods for runtime monitoring of policies should be properly discussed, even if they are not suitable for robot applications.
- Similarly, "Neural Network Repair with Reachability Analysis" should be added to the discussion.
- I encourage the authors to carefully consider whether the structure / presentation can be improved. However, ultimately it is up to the authors to decide for an appropriate structure / taxonomy.

**Audience:**

Yes

**Audience Explanation:**

The survey focuses on the application of formal methods to robot learning, which is a rather specific combination, yet sufficiently broad to justify its scope. The topic is relevant to some individuals in TMLR's audience, in particular by also touching the related and highly relevant field of robot safety.

**Claims And Evidence:**

Yes

**Claims Explanation:**

Reviewer NSyM identified 3 factual errors that, however, have been addressed in the revision.

---

> ### Author Response · Authors · 2026-01-01
> **Confirmation of Requested Changes in Final Submission. Thank you!**
>
> We sincerely thank the area chair and all the reviewers for their thoughtful feedback and constructive suggestions, which have helped us to greatly improve our work. Below, we summarize the primary final revisions to our survey in response to the outlined conditions for acceptance.
>
> > Formal methods for runtime monitoring of policies should be properly discussed, even if they are not suitable for robot applications.
>
> We have now included a complete discussion of run-time monitoring methods divided into two categories. We discuss explicitly specification-based run-time monitoring methods in the first labeled paragraph of Section 4.4., citing the fundamental work by Donzé et al. (2013) and the survey by Bartocci et al. (2018), as well as more recent work by Bakhirkin & Basset (2019), Bonnah & Hoque (2022), Chalupa & Henzinger (2023), Pinisetty et al. (2017), Ferrando & Delzanno (2023), and Henzinger et al. (2025). The previous non-specification-based methods are discussed in the following paragraph.
>
> > "Neural Network Repair with Reachability Analysis" should be added to the discussion.
>
> We have now expanded the discussion of repair methods in the last labeled paragraph of Section 5.3 to include the work from Yang et al. (2022), as well as recent work from others by citing Dong et al. (2021), Usman et al. (2021), Sohn et al. (2023), Xing et al. (2024), Majd et al. (2024), and Tao & Thakur (2025). In the expanded discussion, we clarify how such recent works are most often used to repair neural networks acting as classifiers, generative models, or discrete-action policies with specifications meant for those uses (i.e., specifications constraining the policy's actions), from which we conclude that there remains a promising research direction toward larger scale policy repair.
>
> > I encourage the authors to carefully consider whether the structure / presentation can be improved. However, ultimately it is up to the authors to decide for an appropriate structure / taxonomy.
>
> After careful consideration, we believe that the current structure provides the most-comprehensive and accessible overview of formal methods in robot policy learning and verification possible, and we have not made any major changes to the overall organization of the survey.
>
> Thanks to everyone once again for their valuable feedback!